# UrbanGS: A Scalable and Efficient Architecture for Geometrically Accurate Large-Scene Reconstruction

**Changbai Li**[1,*] **Haodong Zhu**[2,*] **Hanlin Chen**[5] **Xiuping Liang**[2]**, Tongfei Chen**[2]
**Shuwei Shao**[6]**, Linlin Yang**[4,†] **Huobin Tan**[1,3,†] **Baochang Zhang**[2,7]

[1]Hangzhou International Innovation Institute, Beihang University
[2]Institute of Artificial Intelligence, Beihang University
[3]School of Software, Beihang University
[4]State Key Laboratory of Media Convergence and Communication,
  Communication University of China
[5]Department of Computer Science, National University of Singapore
[6]Control Science and Engineering, Shandong University
[7]Artificial Intelligence Research Center, Lobachevsky State University

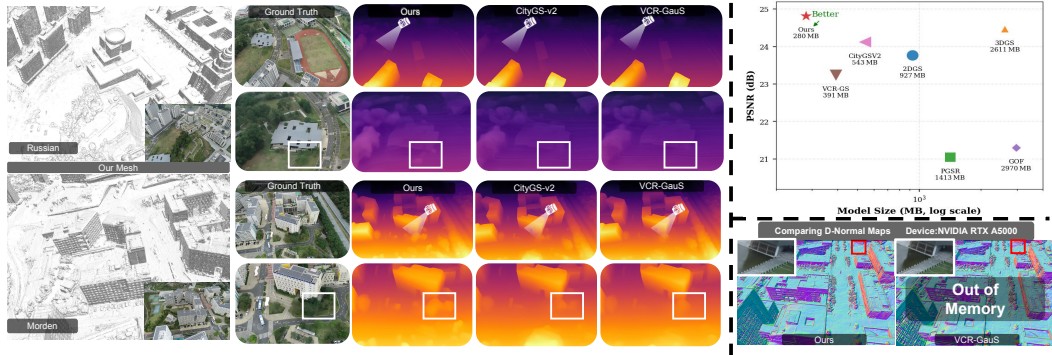

Figure 1: We propose UrbanGS, a scalable framework for high-fidelity large-scale scene reconstruction. Left: It reconstructs complex urban environments from multi-view RGB images, capturing fine details like trees, buildings, and roads. Middle: Compared with CityGS-v2 (Liu et al., 2024b) and VCR-Gaus (Chen et al., 2024b), by comparing rendered depth maps, our method can intuitively demonstrate its geometric advantages in terms of the surface smoothness of objects. Top-right: Our Spatially Adaptive Gaussian Pruning enables significant model compression while preserving quality. Bottom-right: UrbanGS efficiently reconstructs large scenes on A5000 GPUs, whereas VCR-Gaus (Chen et al., 2024b) fails due to out-of-memory issues.

## Abstract

While 3D Gaussian Splatting (3DGS) enables high-quality, real-time rendering for bounded scenes, its extension to large-scale urban environments gives rise to critical challenges in terms of geometric consistency, memory efficiency, and computational scalability. To address these issues, we present UrbanGS, a scalable reconstruction framework that effectively tackles these challenges for city-scale applications. First, we propose a Depth-Consistent D-Normal Regularization module. Unlike existing approaches that rely solely on monocular normal estimators, which can effectively update rotation parameters yet struggle to update position parameters, our method integrates D-Normal constraints with external depth supervision. This allows for comprehensive updates of all geometric parameters. By further incorporating an adaptive confidence weighting mechanism based on

---

[*]Equal contribution.
[†]Corresponding author: lyang@cuc.edu.cn, thbin@buaa.edu.cn

gradient consistency and inverse depth deviation, our approach significantly enhances multi-view depth alignment and geometric coherence, which effectively resolves the issue of geometric accuracy in complex large-scale scenes. To improve scalability, we introduce a Spatially Adaptive Gaussian Pruning (SAGP) strategy, which dynamically adjusts Gaussian density based on local geometric complexity and visibility to reduce redundancy. Additionally, a unified partitioning and view assignment scheme is designed to eliminate boundary artifacts and optimize computational load. Extensive experiments on multiple urban datasets demonstrate that UrbanGS achieves superior performance in rendering quality, geometric accuracy, and memory efficiency, providing a systematic solution for high-fidelity large-scale scene reconstruction.

# 1 INTRODUCTION

3D scene reconstruction is a long-standing research topic in computer vision and computer graphics, with its core objective of achieving photorealistic rendering and accurate geometric reconstruction. Following the introduction of Neural Radiance Fields (NeRF) (Mildenhall et al., 2021), 3D Gaussian Splatting (3DGS) (Kerbl et al., 2023) has emerged as a mainstream technique in this field, thanks to its advantages in training convergence and rendering efficiency. 3DGS represents scenes using a set of discrete Gaussian ellipsoids and leverages a highly optimized rasterizer for rendering. However, due to the unstructured nature of 3DGS, accurately representing surfaces—especially in large-scale complex scenes—remains a significant challenge. In recent years, numerous prominent studies (Huang et al., 2024a; Chen et al., 2024b;a) have been proposed to address this issue. While these methods have achieved remarkable success in single-object or small-scale scene reconstruction, directly extending them to complex large-scale scenes reveals several critical limitations. For instance, vanilla 3DGS suffers from inadequate geometric modeling accuracy and incomplete parameter updates when applied to city-scale environments, failing to meet the high-fidelity reconstruction requirements of complex urban scenes.

To tackle the challenges of urban-scale modeling, various technical solutions have been developed. Methods such as CityGaussian (Liu et al., 2024a) and VastGaussian (Lin et al., 2024) have proposed block-wise partitioning strategies; although these strategies improve rendering efficiency, they still suffer from geometric inconsistencies, low geometric accuracy, and fail to reduce memory requirements during training. CityGaussianv2 (Liu et al., 2024b) adopts a hybrid approach integrating 2D Gaussian Splatting (Huang et al., 2024a), while this accelerates training and enhances geometric accuracy, it comes at the cost of degraded rendering quality. Furthermore, vanilla 3DGS generates excessive redundant Gaussian primitives in homogeneous regions (e.g., skies, distant building facades), and naive pruning heuristics often sacrifice fine-grained details (Fan et al., 2023). Existing partitioning schemes also introduce computational inefficiencies by processing irrelevant views and generating boundary discontinuities (Liu et al., 2024a). These limitations underscore the urgent need for a unified framework that balances geometric precision, memory efficiency, and seamless scalability.

We propose **UrbanGS**, a strategy that achieves high geometric accuracy, fidelity, and efficiency in large-scale scene reconstruction. To enhance geometric fidelity in large-scale settings, we directly supervise the rendered normal maps of 3D Gaussians with external pseudo-normal priors. However, this form of supervision alone is insufficient for updating the position parameters of Gaussians, which is critical for accurate surface reconstruction (Chen et al., 2024b). To overcome this limitation, we introduce a Depth-Consistent D-Normal Regularization framework. Instead of supervising the rendered normals directly, we first derive depth-normal (D-Normal) from the spatial gradient of the rendered depth maps, which are then supervised by the pseudo-normal priors. This establishes a geometric constraint intrinsically linked to depth, thereby enabling comprehensive updates of both rotation and position parameters of the Gaussians. Furthermore, considering the limitation that supervision based on D-Normal relies on the accuracy of rendered depth maps, we introduce a depth estimator (Pseudo Depth) (Hu et al., 2024) to directly supervise the rendered depth maps, thereby constructing the "Pseudo Depth & D-Normal Dual Supervision Mechanism" (with theoretical proofs provided in the supplementary materials). To ensure the reliability of depth alignment across multiple views, we propose an adaptive confidence weighting strategy that dynamically ad-

justs supervision weights for different regions, thus reducing the impact of depth errors on surface reconstruction results.

To meet the memory and computational demands of urban-scale reconstruction, we propose a Spatially Adaptive Gaussian Pruning (SAGP) method. Traditional pruning approaches, designed for small-scale or object-level scenes, rely on global metrics or fixed thresholds (Fan et al., 2023). When applied to city-scale scenes with high spatial heterogeneity and numerous Gaussian primitives, such strategies often oversimplify local structures or lose fine details (see Table 4). To our knowledge, this is the first pruning framework specifically designed for city-scale 3D Gaussian Splatting. SAGP operates within local voxel cells, integrating local geometric complexity, ray-intersection frequency, and visibility-aware importance scores to decide which Gaussians to prune. This adaptively removes redundant primitives—especially in uniform or distant regions—while preserving perceptually and geometrically critical structures. Applied progressively during training, SAGP significantly reduces model complexity and memory usage while maintaining high rendering and geometric quality (Table 4, Fig. 4). We also incorporate a partitioning strategy (Liu et al., 2024a) to enable parallel processing, supporting efficient and scalable reconstruction of large-scale urban scenes.
Our main contributions are summarized below:

- We propose a Depth-Consistent D-Normal Regularizer that enables holistic optimization of all Gaussian parameters (position, rotation), addressing the limitation of incomplete geometric updates in methods that supervise only rendered normals.
- We introduce an adaptive confidence term to enhance robustness, which suppresses unreliable depth predictions and strengthens multi-view geometric alignment.
- To address Gaussian redundancy and memory explosion in city-scale scenes, we design a Spatially Adaptive Gaussian Pruning (SAGP) algorithm that is aware of local geometric complexity.
- Extensive experiments demonstrate that our method outperforms existing large-scale scene reconstruction techniques, thus laying a solid foundation for future further research in this field.

## 2    RELATED WORK

**Neural Rendering.** Novel view synthesis and multi-view surface reconstruction are interconnected tasks in 3D scene reconstruction. Traditional reconstruction pipelines relied on Structure-from-Motion (SfM) (Duisterhof et al., 2024; Wang et al., 2023; He et al., 2024) and Multi-View Stereo (MVS) (Furukawa et al., 2015; Tang et al., 2024b; Yao et al., 2018) with feature matching (Wang & Shen, 2018), but suffered from artifacts and noise sensitivity (Leroy et al., 2024). Early synthesis methods like Soft3D (Penner & Zhang, 2017) used volumetric ray-marching with high computational costs. The neural revolution began with NeRF (Mildenhall et al., 2021), which improved quality through positional encoding yet remained slow due to MLPs; variants like Mip-NeRF (Barron et al., 2022), InstantNGP (Müller et al., 2022), and Plenoxels (Fridovich-Keil et al., 2022) balanced efficiency but struggled with empty spaces. For reconstruction, implicit methods like NeuS (Wang et al., 2021) and Neuralangelo (Li et al., 2023) integrated SDFs (Park et al., 2019; Yu et al., 2022) for detailed surfaces at the cost of lengthy training. The paradigm shifted with 3D Gaussian Splatting (3DGS) (Kerbl et al., 2023), enabling real-time synthesis via unstructured Gaussians, though its explicit form caused reconstruction issues like depth ambiguities (Zhang et al., 2024; Chen et al., 2024a). Subsequent optimizations addressed both domains: synthesis-focused improvements included Mip-splatting (Yu et al., 2024a), while reconstruction enhancements featured SuGaR's mesh binding (Guédon & Lepetit, 2024) (despite scalability limits (Chen et al., 2024a)), 2DGS's surfel-based normal alignment (Huang et al., 2024b), VCR-GauS's depth-normal regularizers (Chen et al., 2024b), and GOF's ray-tracing for unbounded scenes (Yu et al., 2024b). However, when reconstructing complex large-scale scenes, 3DGS faces considerable challenges in terms of rendering quality and geometric accuracy. Furthermore, it also has the problems of a surge in video memory usage and excessively long training times, all of which limit the further expansion of 3DGS in large-scale scenes.

**Large-Scale Scene Reconstruction.** Reconstructing large-scale scenes (e.g., urban areas, expansive landscapes) faces significant challenges, including computational inefficiency, memory constraints, and geometric inconsistencies across sub-scenes processed in a block-wise manner (Tancik et al., 2022). Early NeRF-based methods partitioned scenes into blocks for parallel training (Turki et al., 2022; Zhang et al., 2025), but due to the limitations of multi-layer perceptrons (MLPs), these approaches suffered from slow rendering speeds and poor scalability (Kerbl et al., 2023). Although

3DGS-based methods improved efficiency, they introduced new issues: partition-and-merge strategies such as VastGaussian (Lin et al., 2024) often lead to boundary inconsistencies due to insufficient multi-view constraints; methods like CityGaussian (Liu et al., 2024a) require time-consuming post-processing for pruning or distillation; and while these methods improve rendering quality, they still struggle with geometric accuracy, training cost, and efficiency (Chen & Lee, 2024). More recently, CityGS-X (Gao et al., 2025) revisits large-scale 3DGS from a systems perspective, introducing a parallel hierarchical representation with multi-task supervision and progressive optimization that eliminates the partition-and-merge pipeline and improves geometric consistency under scalable multi-GPU training, but its surface reconstruction quality for high-fidelity urban details remains limited. Optimization-focused solutions like CityGaussianV2 (Liu et al., 2024b), despite employing techniques to control Gaussian proliferation, sacrifice rendering quality to some extent. To address these limitations, we propose the UrbanGS framework, which establishes a unified depth-normal regularizer for holistic geometric optimization, incorporates confidence-aware weighting to enhance robustness, introduces spatially adaptive pruning to manage redundancy, and designs a seamless partitioning scheme, collectively achieving high-fidelity, efficient, and geometrically consistent large-scale reconstruction.

## 3 METHODOLOGY

### 3.1 PRELIMINARIES

**3D Gaussian Splatting.** 3D Gaussian Splatting models a scene using a collection of anisotropic 3D Gaussians $G = \{G_i \mid i \in \mathbb{N}\}$. Each 3D Gaussian unit $G_i$ is characterized by a center $u \in \mathbb{R}^3$ and a covariance matrix $\Sigma \in \mathbb{R}^{3 \times 3}$, and can be mathematically expressed as:

$$G_i(p) = \exp\left\{ -\frac{1}{2} \left(p - u_i\right)^\top \Sigma_i^{-1} \left(p - u_i\right) \right\}. \tag{1}$$

During the training process, the covariance matrix is decomposed into a rotation matrix $R \in \mathbb{R}^{3 \times 3}$ and a diagonal scaling matrix $S \in \mathbb{R}^{3 \times 3}$, that is,

$$\Sigma_i = RSS^\top R^\top, \tag{2}$$

to ensure the covariance matrix is positive semi-definite. For rendering the color of a pixel $p$, the 3D Gaussians are projected into the image space for alpha blending:

$$C = \sum_i c_i \alpha_i \prod_{j=1}^{i-1} \left(1 - \alpha_j\right), \tag{3}$$

where $c_i$ and $\alpha_i = o_i G(x_i)$ denote the color and density of a point, respectively.

### 3.2 DEPTH-CONSISTENT D-NORMAL REGULARIZATION

**D-Normal Regularization.** To reconstruct scene surfaces, we enforce normal priors $N$ predicted by a pretrained monocular deep neural network (Bae & Davison, 2024) to supervise the rendered normal map $\hat{N}$ using $L_1$ and cosine losses:

$$\mathcal{L}_n = ||\hat{N} - N||_1 + (1 - \hat{N} \cdot N). \tag{4}$$

In our method, the depth map is rendered by performing a weighted sum of depths (Bae & Davison, 2024; Chen et al., 2024b; Yu et al., 2022), with the formula given as follows:

$$\hat{D} = \frac{\sum_{i \in M} d_i \alpha_i \prod_{j=1}^{i-1}(1 - \alpha_j)}{\sum_{i \in M} \alpha_i \prod_{j=1}^{i-1}(1 - \alpha_j)}, \tag{5}$$

where $d_i$ denotes the intersection depth (Chen et al., 2023; 2024b) and is distinct from the depth estimation in conventional 3D Gaussian Splatting (3DGS). Specifically, it refers to the distance from the camera to the intersection point calculated along the z-axis of the camera coordinate system; this

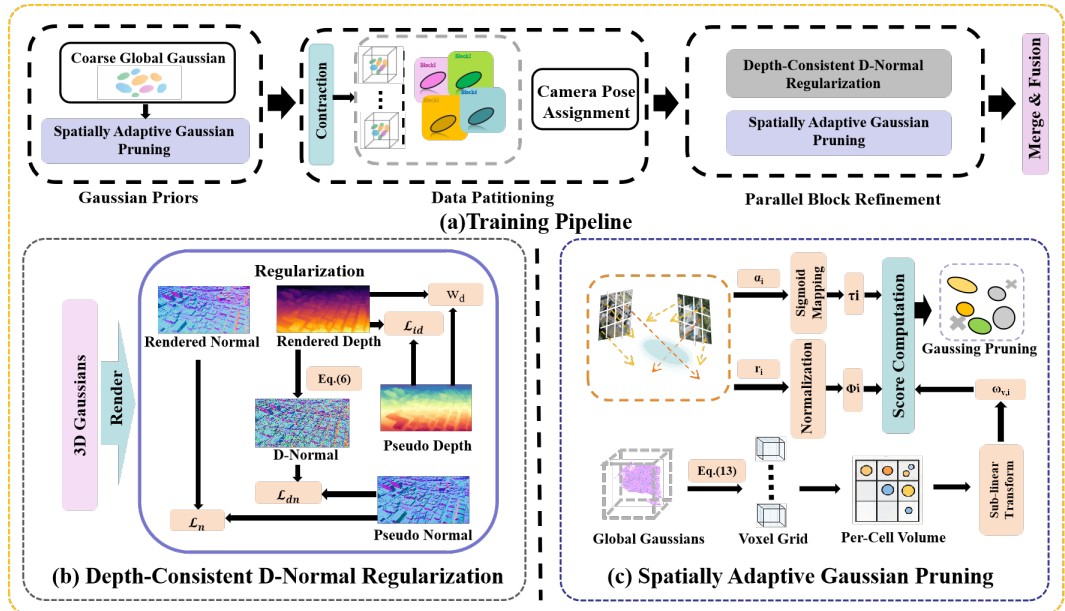

Figure 2: **UrbanGS training pipeline and core components.** (a) **Training Pipeline:** Starting from coarse global Gaussians, we apply spatially adaptive Gaussian pruning to obtain compact priors, contract and partition the scene into blocks, assign camera views using geometric and SSIM-based criteria, and refine all blocks in parallel before merging them into a unified large-scale 3D Gaussian scene. (b) **Depth-Consistent D-Normal Regularization:** 3D Gaussians are rendered to depth and normal maps, depth is converted to D-normals and jointly supervised with pseudo-depth and pseudo-normal priors from pretrained models via the loss $\mathcal{L}_n + \mathcal{L}_{dn} + w_d\mathcal{L}_{id}$, yielding stable and globally consistent geometry. (c) **Spatially Adaptive Gaussian Pruning:** Global Gaussians are discretized into a voxel grid, where per-cell importance $\omega_{v,i}$ and view-dependent cues are fused into pruning scores to remove redundant Gaussians and obtain an efficient yet accurate representation.

intersection point is formed between the ray emitted from the camera center and the elliptical plane obtained by compressing the ellipsoid of 3DGS (Further details are provided in the supplementary material in the Appendix).

Additionally, to effectively update Gaussian positions, we utilize the predicted normal $N$ from the pretrained model to supervise the D-Normal $\overline{N}_d$. The derivation of the D-Normal from the rendered depth involves two sequential steps. First, the rendered depth map is back-projected into point clouds $\{\mathbf{d}_k(n,p)\}$, using the camera intrinsic matrix. Subsequently, the horizontal and vertical finite differences are computed between adjacent points in this back-projected point cloud; the D-Normal is then obtained by calculating the cross-product of these two sets of finite differences.

$$\overline{N}_d(n,p) = \frac{\nabla_v d(n,p) \times \nabla_h d(n,p)}{|\nabla_v d \times \nabla_h d|}, \tag{6}$$

where $d$ represents the 3D coordinates of a pixel obtained via back-projection from the depth map. We then apply the D-Normal regularization:

$$\mathcal{L}_{dn} = \left\|\overline{N}_d - N\right\|_1 + \left(1 - \overline{N}_d \cdot N\right), \tag{7}$$

**Depth Consistency Regularization.** In urban-scale scenes, D-Normal regularization optimizes geometry through normal-depth associations but lacks explicit cross-view depth constraints, frequently causing building misalignment and street distortion—especially in distant/complex areas. To resolve inconsistent multi-view depth predictions, we propose a depth consistency framework integrating inverse depth constraints with geometry-aware confidence. This extends normal-based regularization by incorporating robust priors from monocular depth estimators, where depth anchors $D_{\text{ext}}$(Hu et al., 2024) ensure cross-view consistency during optimization.

Specifically, we derive dense relative depth anchors by processing training images with a pre-trained DepthAnything-v2 model (Hu et al., 2024). To align these monocular predictions with the unified metric scale of the 3D reconstruction, we leverage sparse 3D points from COLMAP's Structure-from-Motion (SfM) (Schönberger & Frahm, 2016). Specifically, we compute per-view scale and shift parameters by robustly fitting the monocular depth maps to the sparse COLMAP depth values at valid 2D-3D correspondences. This process brings the relative depth estimates into alignment with the scale of the multi-view geometry. We define an inverse depth loss $\mathcal{L}_{\mathrm{id}}$ that operates on reciprocal depths to balance optimization sensitivity across distance ranges (Kerbl et al., 2024):

$$\mathcal{L}_{\mathrm{id}}(u,v) = \left| \hat{D}^{-1}(u,v) - D_{\mathrm{ext}}^{-1}(u,v) \right|. \tag{8}$$

where $\hat{D}^{-1} \equiv 1/\hat{D}$ is the reciprocal of the rendered depth map. This formulation minimizes relative depth errors per pixel while enhancing distant surface accuracy where linear depth gradients diminish. Complementing this loss, we define a geometry-aware confidence measure $w_d$ based on two geometric cues. First, the cosine similarity of depth gradients:

$$\cos\phi = \frac{\nabla\hat{D} \cdot \nabla D_{\mathrm{ext}}}{\|\nabla\hat{D}\|_2 \|\nabla D_{\mathrm{ext}}\|_2}, \tag{9}$$

quantifies gradient reliability by measuring local surface orientation consistency. Second, we measure error sensitivity via normalized inverse depth deviation to suppress high-discrepancy regions:

$$\epsilon_d(u,v) = \frac{\mathcal{L}_{\mathrm{id}}(u,v)}{\mathrm{median}(\hat{D}^{-1})}. \tag{10}$$

The unified confidence $w_d$ combines both cues through exponential decay:

$$w_d = \exp\left(\frac{\cos\phi - 1}{0.01}\right) \cdot \exp\left(-\frac{\epsilon_d}{0.1}\right), \tag{11}$$

where hyperparameters $\gamma_d = 0.01$ and $\tau = 0.1$ balance directional and magnitude sensitivity. The total optimization objective is consequently augmented to:

$$\mathcal{L}_{\mathrm{total}} = \mathcal{L}_{\mathrm{RGB}} + \lambda_1 \mathcal{L}_{\mathrm{n}} + \lambda_2 \mathcal{L}_{\mathrm{dn}} + \lambda_3 (w_d \cdot \mathcal{L}_{\mathrm{id}}), \tag{12}$$

where $\lambda_i (i = 1, 2, 3)$ balancing the individual components. $\mathcal{L}_{\mathrm{RGB}}$ includes $\mathcal{L}_1$ and D-SSIM losses (Kerbl et al., 2023).

### 3.3 SPATIALLY ADAPTIVE GAUSSIAN PRUNING (SAGP)

Large-scale 3D scenes exhibit strong spatial heterogeneity: detailed foreground regions require dense Gaussians to capture fine structures, whereas distant areas often suffer from excessive Gaussian proliferation, leading to high memory cost and degraded rendering. Existing pruning strategies based on global metrics or fixed opacity thresholds (Kerbl et al., 2023; Fan et al., 2023) tend to oversimplify local details or remove important far-field Gaussians, resulting in incomplete reconstructions and visual artifacts.

To overcome these limitations, we propose a unified, spatially adaptive pruning framework. The scene is first partitioned into volumetric cells whose characteristic length $\ell$ scales with the overall Gaussian density:

$$\ell = \lambda \left(\frac{\mathcal{V}_{\mathrm{scene}}}{\mathcal{N}}\right)^{1/3}, \tag{13}$$

where $\mathcal{V}_{\mathrm{scene}}$ denotes the bounding-box volume and $\mathcal{N}$ the total number of Gaussians. We set $\lambda = 1.2$ to slightly enlarge the cell size for more stable local statistics.

Within each cell, we compute the $t$-th percentile Gaussian volume $\vartheta_{\mathrm{local}}^{(t)}$ and normalize individual volumes via a sub-linear transform:

$$w_{v,i} = \left(\min\left(\frac{v_i}{\vartheta_{\mathrm{local}}^{(t)}}, 1\right)\right)^{\kappa}. \tag{14}$$

We use $t = 90\%$ to represent the typical volume in each cell while mitigating outlier influence. The sub-linear exponent $\kappa = 0.5$ (i.e., a square root) is applied to compress the dynamic range of volume ratios, thereby amplifying the importance of fine-scale structures while suppressing overly large Gaussians. This operation attenuates oversized background Gaussians while amplifying fine-scale structures, thereby establishing a context-aware basis for importance estimation. Building on these localized volume weights, we define the importance score $S_i$ for each Gaussian as the product of three normalized geometric and photometric attributes: the normalized ray-intersection frequency $\phi_i = \frac{r_i}{\max_{j \in \mathcal{G}(i)} r_j}$, where $r_i$ counts the intersections between the $i$-th Gaussian and sampled rays during training; the Sigmoid-mapped opacity $\tau_i = \sigma(a_i) = \frac{1}{1 + e^{-a_i}}$ derived from the learnable opacity parameter $a_i$; and the sub-linear volume weight $w_{v,i}$ from Eq. 14. The combined score is given by

$$S_i = \phi_i \cdot \tau_i \cdot w_{v,i}, \tag{15}$$

which eliminates the need for manually tuned weighting hyperparameters—a detailed discussion of linear combinations is provided in the supplementary material(Sec. D.5) and ensures that a Gaussian is retained only when it simultaneously exhibits high visibility, frequent observation across views, and appropriate geometric scale.

### 3.4 PARTITIONING STRATEGY

Our partitioning strategy is improved based on CityGS (Liu et al., 2024a), as illustrated in part (a) of Fig. 2. First, when obtaining the global coarse 3DGS model, we first eliminate redundant Gaussians through SAGP pruning to prevent these redundant Gaussians from attracting non-contributing views and amplifying the computational load during subsequent block-wise training. Then, in the partitioning phase, we retain common Gaussian primitives at the boundaries of each sub-block to avoid introducing visible fusion artifacts caused by geometric discontinuities between blocks. All other modules follow the methodologies of CityGS, and the specific formulas are referred to in the supplementary materials C.

## 4 EXPERIMENTS

Table 1: Quantitative comparisons on the Mill19 (Yu et al., 2022) and UrbanScene3D (Lin et al., 2022) datasets for novel view synthesis. ↑ indicates higher is better, while ↓ indicates lower is better. The top three results are highlighted with red, orange, and yellow backgrounds, respectively. $^{\dagger}$ denotes results obtained without the decoupled appearance encoding.

| | Building | | | Rubble | | | Residence | | | Sci-Art | | |
|---|---|---|---|---|---|---|---|---|---|---|---|---|
| | SSIM ↑ | PSNR ↑ | LPIPS ↓ | SSIM ↑ | PSNR ↑ | LPIPS ↓ | SSIM ↑ | PSNR ↑ | LPIPS ↓ | SSIM ↑ | PSNR ↑ | LPIPS ↓ |
| **w/o Geometric Optimization** | | | | | | | | | | | | |
| Mega-NeRF | 0.547 | 20.92 | 0.454 | 0.553 | 24.06 | 0.508 | 0.628 | 22.08 | 0.401 | 0.770 | 25.60 | 0.312 |
| Switch-NeRF | 0.579 | 21.54 | 0.397 | 0.562 | 24.31 | 0.478 | 0.654 | 22.57 | 0.352 | 0.795 | 26.51 | 0.271 |
| VastGaussian † | 0.728 | 21.80 | 0.225 | 0.742 | 25.20 | 0.264 | 0.699 | 21.01 | 0.261 | 0.761 | 22.64 | 0.261 |
| 3DGS | 0.738 | 22.53 | 0.214 | 0.725 | 25.51 | 0.316 | 0.745 | 22.36 | 0.247 | 0.791 | 24.13 | 0.262 |
| DoGaussian | 0.759 | 22.73 | 0.204 | 0.765 | 25.78 | 0.257 | 0.740 | 21.94 | 0.244 | 0.804 | 24.42 | 0.219 |
| CityGaussian | 0.778 | 21.55 | 0.246 | 0.813 | 25.77 | 0.228 | 0.813 | 22.00 | 0.211 | 0.837 | 21.39 | 0.230 |
| **w/ Geometric Optimization** | | | | | | | | | | | | |
| SuGaR | 0.507 | 17.76 | 0.455 | 0.577 | 20.69 | 0.453 | 0.603 | 18.74 | 0.406 | 0.698 | 18.60 | 0.349 |
| NeuS | 0.463 | 18.01 | 0.611 | 0.480 | 20.46 | 0.618 | 0.503 | 17.85 | 0.533 | 0.633 | 18.62 | 0.472 |
| Neuralangelo | 0.582 | 17.89 | 0.322 | 0.625 | 20.18 | 0.314 | 0.644 | 18.03 | 0.263 | 0.769 | 19.10 | 0.231 |
| PGSR | 0.480 | 16.12 | 0.573 | 0.728 | 23.09 | 0.334 | 0.746 | 20.57 | 0.289 | 0.799 | 19.72 | 0.275 |
| VCR-Gaus | 0.502 | 19.56 | 0.502 | 0.541 | 21.34 | 0.428 | 0.623 | 20.59 | 0.359 | 0.665 | 19.31 | 0.465 |
| CityGaussianV2 | 0.650 | 19.07 | 0.397 | 0.720 | 23.75 | 0.322 | 0.769 | 21.15 | 0.234 | 0.810 | 20.66 | 0.266 |
| Ours | 0.802 | 22.82 | 0.208 | 0.791 | 26.25 | 0.210 | 0.823 | 22.48 | 0.205 | 0.824 | 22.62 | 0.279 |

### 4.1 EXPERIMENTAL SETUP

Our experiments cover seven representative scenes drawn from four datasets: Building and Rubble from Mill-19 (Yu et al., 2022); Residence and Sci-Art from UrbanScene3D (Lin et al., 2022); and Residence, Russian Building, and Modern Building from GauU-Scene (Xiong et al., 2024). Unless otherwise noted, competing methods were evaluated on RTXA800 GPUs, while UrbanGS was trained on eight RTXA5000 GPUs. Additional details on training protocols and evaluation settings are provided in the supplementary material.

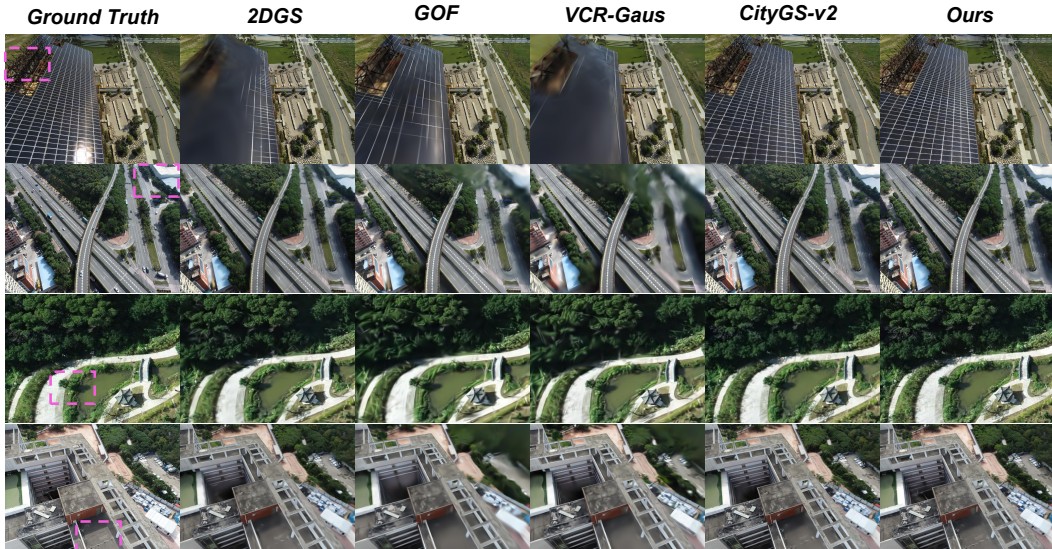

Figure 3: Qualitative results of ours and other methods in image rendering on Mill-19 (Yu et al., 2022) and Urbanscene3D (Lin et al., 2022).

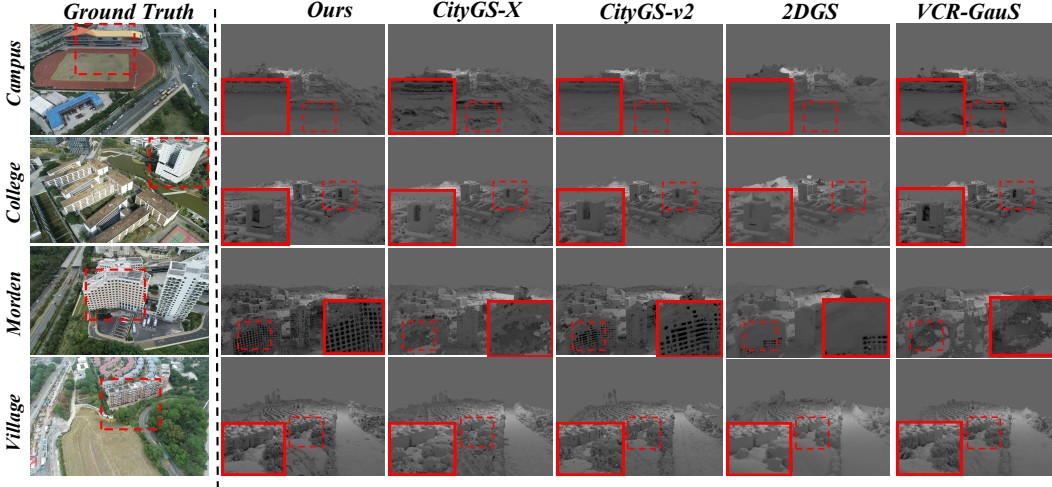

Figure 4: Qualitative mesh and texture comparison between SOTA and our method on GauU-Scene dataset (Xiong et al., 2024).

## 4.2 MAIN RESULTS

**Novel View Synthesis**. As shown in Table 1 and Fig. 3, we present quantitative and qualitative evaluations of large-scale scene reconstruction methods with and without geometric optimization. UrbanGS consistently achieves state-of-the-art performance, attaining the highest PSNR and SSIM in building scenes and reducing LPIPS by 0.006 over CityGS (Liu et al., 2024a) in residential scenes. Qualitative results in Fig. 3 further show reduced floating artifacts, indicating stronger multi-view consistency and more faithful appearance preservation for large-scale reconstruction.

**Surface Reconstruction**. We compare our method with existing surface reconstruction approaches on the GauU-Scene datasets (Xiong et al., 2024). As shown in Table 2, our method achieves state-of-the-art performance among both neural implicit baselines and recent 3DGS-based city-scale methods. In particular, compared with CityGS-X, our approach attains higher F1 scores across all scenes by improving recall while maintaining comparable precision. It also surpasses CityGS-v2 (Liu et al.,

Table 2: Detailed geometry evaluation on the GauU-Scene dataset (Xiong et al., 2024). "NaN" indicates that the method produced invalid numerical results, while "FAIL" denotes a failure to extract a valid mesh. For all metrics, ↑ indicates that higher values are better.

| Methods | Residence | | | Russian Building | | | Modern Building | | |
|---|---|---|---|---|---|---|---|---|---|
| | P ↑ | R ↑ | F1 ↑ | P ↑ | R ↑ | F1 ↑ | P ↑ | R ↑ | F1 ↑ |
| NeuS | FAIL | FAIL | FAIL | FAIL | FAIL | FAIL | FAIL | FAIL | FAIL |
| Neuralangelo | NaN | NaN | NaN | FAIL | FAIL | FAIL | NaN | NaN | NaN |
| SuGaR | 0.579 | 0.287 | 0.384 | 0.480 | 0.369 | 0.417 | 0.650 | 0.220 | 0.329 |
| GOF | 0.404 | 0.418 | 0.411 | 0.294 | 0.394 | 0.330 | 0.411 | 0.357 | 0.382 |
| VCR-Gaus | 0.498 | 0.402 | 0.445 | 0.538 | 0.454 | 0.492 | 0.591 | 0.401 | 0.478 |
| 2DGS | 0.526 | 0.406 | 0.458 | 0.544 | 0.519 | 0.531 | 0.588 | 0.413 | 0.485 |
| CityGS-X | 0.512 | 0.411 | 0.456 | 0.572 | 0.516 | 0.542 | 0.653 | 0.389 | 0.487 |
| CityGaussianV2 | 0.524 | 0.421 | 0.467 | 0.560 | 0.530 | 0.544 | 0.643 | 0.398 | 0.492 |
| Ours | 0.529 | 0.461 | 0.493 | 0.568 | 0.525 | 0.546 | 0.662 | 0.408 | 0.503 |

Table 3: Under the GauU-Scene dataset (Lin et al., 2022), comparison of Large-Scale Scene Modeling Methods, the best result for specific metrics under each scene is highlighted in **bold**.

| Scene | Method | PSNR ↑ | F1 ↑ | #GS(M) ↓ | Size(G) ↓ | Mem.(G) ↓ |
|---|---|---|---|---|---|---|
| Residence | CityGS | 23.17 | 0.453 | 8.05 | 0.44 | 31.5 |
| | CityGS-v2 | 23.46 | 0.465 | 8.07 | 0.44 | 14.2 |
| | Ours | **23.78** | **0.493** | **7.78** | **0.37** | **13.2** |
| Russia | CityGS | 24.19 | 0.455 | 7.00 | 0.38 | 27.4 |
| | CityGS-v2 | 23.89 | 0.537 | 6.97 | 0.38 | 15.0 |
| | Ours | **24.53** | **0.546** | **6.56** | **0.35** | **11.4** |
| Modern | CityGS | 26.22 | 0.462 | 7.90 | 0.43 | 29.2 |
| | CityGS-v2 | 25.53 | 0.489 | 7.90 | 0.42 | 16.1 |
| | Ours | **26.44** | **0.503** | **7.45** | **0.39** | **15.0** |

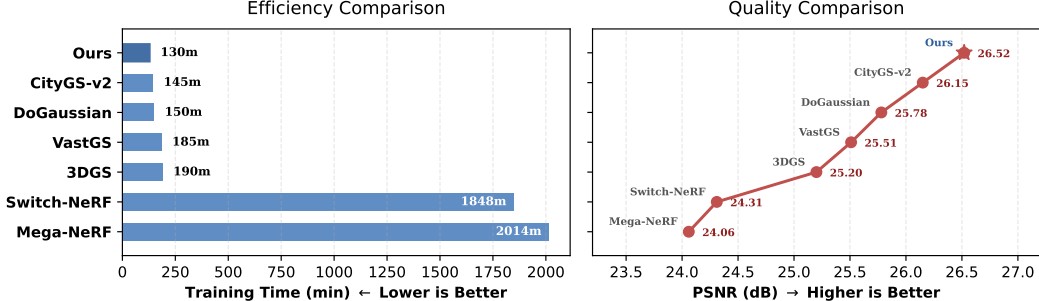

Figure 5: Experimental results on the Rubble dataset (Yu et al., 2022) demonstrate that the proposed method outperforms comparative approaches in terms of PSNR while achieving superior training efficiency.

2024b) on most metrics. Qualitative comparisons in Figure 4 further show that our method produces more detailed and clearer surface structures. Additional mesh visualizations are provided in Appendix B.

**Efficiency Comparison.** We compare the training time of our method with that of existing methods. As shown in Fig. 5, our method only takes 2 hours and 10 minutes to complete the training on the Rubble (Lin et al., 2022), which is significantly faster than competing methods. As presented in Table. 3, when compared with other large-scale scene algorithms, our method requires lower computational costs while achieving better rendering quality and geometric accuracy.

Table 4: Ablation Results on Russian dataset (Xiong et al., 2024). **Bold** indicates best performance. Note that OOM denotes Out Of Memory.

| Method | Rendering Quality | | | Geometric Quality | | | Training Statistics | | | |
|---|---|---|---|---|---|---|---|---|---|---|
| | PSNR↑ | SSIM↑ | LPIPS↓ | P↑ | R↑ | F1↑ | GS (M)↓ | Time↓ | Size↓ | Mem↓ |
| Baseline | 22.54 | 0.778 | 0.231 | 0.532 | 0.501 | 0.516 | 6.43 | 235 | 1102.23 | OOM |
| +ST | **24.68** | **0.816** | 0.188 | **0.571** | 0.518 | 0.543 | 6.37 | 188 | 1035.02 | 26.3 |
| +LP | 24.53 | 0.785 | 0.195 | 0.556 | 0.502 | 0.528 | 3.02 | 134 | 467.47 | 17.1 |
| +SAPG (Ours) | 24.66 | 0.813 | **0.184** | 0.568 | **0.525** | **0.546** | **2.45** | 122 | **314.24** | 14.4 |
| STPG | 24.57 | 0.801 | 0.201 | 0.563 | 0.511 | 0.536 | 2.73 | **119** | 320.12 | **13.9** |

Table 5: Ablation study on the effects of D-Normal Regularization and Depth Consistency Regularization, conducted on the Morden Building dataset (Xiong et al., 2024). **Bold** indicates best performance.

| Method | PSNR↑ | SSIM↑ | LPIPS↓ | F1↑ |
|---|---|---|---|---|
| w/o D-Normal | 25.02 | 0.743 | 0.215 | 0.463 |
| w/o Depth Consistency | 24.59 | 0.792 | 0.201 | 0.453 |
| w/o Geometry-Aware Confidence | 26.02 | 0.795 | 0.163 | 0.493 |
| Full | **26.44** | **0.805** | **0.157** | **0.503** |

## 4.3 ABLATION STUDIES

To validate the effectiveness of individual components in our method, we conduct a series of ablation studies on the GauU-Scene dataset. Specifically, we evaluate the impact of the following components: Spatially Adaptive Gaussian Pruning (SAGP), Depth-Consistent D-Normal Regularization, and the partitioning strategy.

**Ablation of SAGP & Gaussian Partitioning .** As summarized in Tab. 4, we conduct a systematic ablation to evaluate the individual contributions of our proposed SAGP and partitioning strategy. We first establish a Baseline that employs neither our SAGP nor any partitioning strategy.

For SAGP, we compare our SAGP pruning against LP, the pruning method from LightGaussian (Fan et al., 2023). The results demonstrate that our SAGP is more effective at preserving the original geometric quality (higher F1 score) while significantly reducing the number of Gaussians, training time, and memory consumption, with only a minor impact on rendering quality.

For Partitioning Strategy, our full method (Ours) integrates the proposed partitioning strategy (ST) with SAGP. We further compare it against STPG, which uses the partitioning strategy from City-Gaussian (Liu et al., 2024a) with our SAGP. The comparison validates the superior effectiveness of our partitioning strategy, as it achieves better rendering and geometric quality under the same pruning method, demonstrating its ability to better preserve structural consistency across blocks.

**Ablation of Depth-Consistent D-Normal Regularization.** As shown in Tab. 5, we conduct ablation studies on each component of the Depth-Consistent D-Normal Regularization, demonstrating that its introduction significantly enhances both rendering quality and geometric accuracy for large-scale scenes. Quantitative results reveal consistent improvements across all evaluation metrics, with notable gains in F1-score (from 0.453 to 0.503) and PSNR (from 24.59 to 26.44), validating the critical importance of this component for high-quality large-scale reconstruction. Furthermore, as illustrated in Fig. D, the Geometric Regularization substantially improves the details in rendered images, as well as the quality of rendered normal and depth maps.

## 5 CONCLUSIONS

This paper presents UrbanGS, a scalable framework for urban-scale scene reconstruction. It introduces a depth-consistent D-Normal regularizer that enables comprehensive optimization of all Gaussian geometric parameters by fusing depth and normal cues. A spatial pruning strategy and seamless partitioning further enhance efficiency and avoid artifacts. Experiments show UrbanGS outperforms existing methods in rendering, geometry, and training speed, offering a practical solution for large-scale 3D reconstruction.

ACKNOWLEDGEMENTS

This work was supported by the National Key R&D Program of China (Grant No. 2024YFC3308200), the Key R&D Program of Henan Province of China (Grant No. 231111211500), and the National Natural Science Foundation of China (Grant No. 62406298). The experiments part was supported by the Ministry of Economic Development of the Russian Federation (agreement identifier 000000C313925P3X0002, Grant No. 139-15-2025-004 dated April 17, 2025).

ETHICS STATEMENT

This work presents UrbanGS, a scalable framework for high-fidelity large-scale scene reconstruction. The research focuses on methodological innovation to address challenges in geometric consistency, memory efficiency, and computational scalability of 3D Gaussian Splatting in urban-scale applications. All experiments use publicly available benchmark datasets (Mill-19, UrbanScene3D, GauU-Scene) in line with academic practices, involving no human subjects, personal data, or social risk assessment. The authors encourage ethical and legal use of this technology and declare no potential conflicts of interest.

REPRODUCIBILITY STATEMENT

To ensure reproducibility of UrbanGS's results, we provide key details: the proposed components (Depth-Consistent D-Normal Regularization, Spatially Adaptive Gaussian Pruning, partitioning strategy) are detailed in the methodology section with mathematical formulations . Experimental setups include training on 8 NVIDIA RTX A5000 GPUs (baselines on RTX A800), using PyTorch 2.0+, Open3D 0.18.0+, and pretrained models (DepthAnything-v2, Dsine) . Dataset preparation follows the image downsampling strategy (resizing images wider than 1600 pixels) and original train/validation splits . We will make the complete code and training scripts publicly available on GitHub upon the final revision and acceptance of this paper.

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

# A    IMPLEMENTATION DETAILS

**Training Setup.** UrbanGS are trained using NVIDIA A5000 GPUs, while all baseline methods are trained on NVIDIA A800 GPUs. Since the Mill-19 (Yu et al., 2022), UrbanScene3D (Lin et al., 2022), and GauU-Scene (Xiong et al., 2024) datasets contain thousands of high-resolution images, we follow the image downsampling strategy proposed in 3DGS: any image with a width exceeding 1600 pixels is resized proportionally during both training and validation.For geometric priors, we utilize the DepthAnything-v2 model (Hu et al., 2024) for depth prediction and the pre-trained Dsine model (Bae & Davison, 2024) for surface normal estimation.

Regarding the pruning schedule, our design follows the training dynamics of 3DGS and prior practice. As shown in the pipeline 2, we use two stages of pruning. When constructing the coarse global Gaussian model, we apply an initial, simple pruning rule to remove obviously redundant Gaussians, reduce memory, and obtain a compact global prior for subsequent block-wise training. During block refinement, we prune at 7k, 15k, and 25k iterations (out of 30k). The 7k step is applied after the scene has roughly formed and the Gaussian distribution starts to stabilize, consistent with the behavior observed in 3DGS (Kerbl et al., 2023), and removes early exploratory Gaussians that no longer contribute to the final geometry. The 15k step follows the original 3DGS setting, occurring at the end of densification when the Gaussian count peaks, and is most effective for controlling model complexity and overfitting. The final pruning at 25k, inspired by LightGaussian (Fan et al., 2023), acts as a consolidation step near convergence, further eliminating residual redundancy and ensuring a good balance between high-fidelity reconstruction and compact, efficient rendering.

**Mesh Extraction.** To obtain the final mesh, we employ Open3D's volumetric TSDF fusion method, which integrates rendered depth maps and corresponding camera poses to construct a continuous Signed Distance Field (SDF). The surface is then extracted using the Marching Cubes algorithm at the zero-level isosurface, enabling direct reconstruction of 3D geometry without relying on intermediate point cloud representations.

# B    PROOF ON A DEPTH-CONSISTENT D-NORMAL REGULARIZER

These propositions systematically validate the evolutionary process from traditional rendered normal supervision to our proposed depth-normal regularizer. Proposition 1.1 reveals the limitation of supervising only rendered normals in updating Gaussian positions; Proposition 1.2 demonstrates that the depth-normal regularizer can effectively optimize Gaussian positions; Proposition 2.1 further proves that the depth-consistent regularizer significantly improves geometric accuracy, highlighting the enhanced effectiveness of our method.

## B.1    GEOMETRIC PROPERTIES

To reconstruct the 3D surface, we focus on the geometric properties of Gaussians that enable accurate intersection depth calculation, as detailed below.

**Normal Vector** Following NeuSG (Chen et al., 2023), the Gaussian's normal vector $\mathbf{n} \in \mathbb{R}^3$ is defined as the direction of its minimized scaling factor:

$$\mathbf{n} = \mathbf{R}[k, :], \quad k = \arg \min \left( [s_1, s_2, s_3] \right). \tag{16}$$

Both $\mathbf{n}$ and Gaussian center $\mathbf{p}$ are transformed to the camera coordinate system (default unless stated otherwise).

**Intersection Depth** Existing work (Tang et al., 2024a) uses $\mathbf{p}$ for depth calculation, which is inaccurate (depth unrelated to $\mathbf{n}$). We instead compute the ray-Gaussian intersection depth via the following steps:

Gaussian Flattening with Scale Regularization: To simplify intersection computation, we adopt NeuSG's (Chen et al., 2023) scale loss to flatten 3D Gaussian ellipsoids into planes $(\mathbf{p}, \mathbf{n})$:

$$\mathcal{L}_s = \|\min \left( s_1, s_2, s_3 \right)\|_1. \tag{17}$$

This loss constrains the minimum scaling factor component to approach zero.

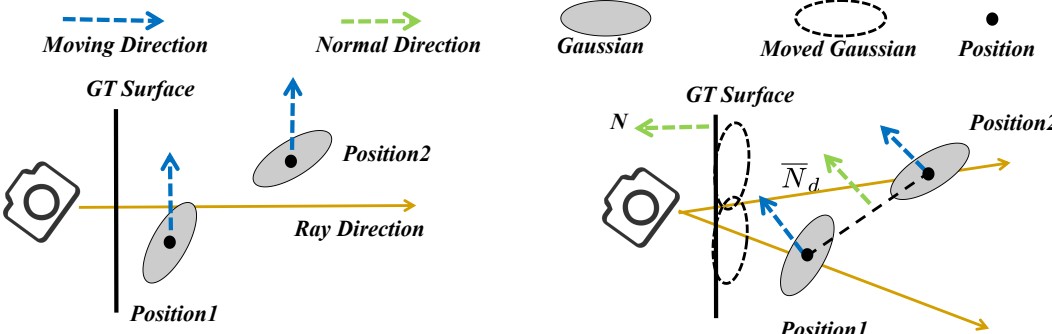

Figure A: Illustration of Proof of the Proposition on Comprehensive Update of Gaussian Parameters. (a) After back-propagation through alpha-blending Eq. 1, the rendered normal supervision loss $\mathcal{L}_n$ moves Gaussians either closer to (corresponding to $Position_1$) or farther from (corresponding to $Position_2$) the intersecting ray. When the normal of a Gaussian is closer to the ground truth (GT) surface normal, this supervision mechanism pushes the Gaussian (e.g., $Position_1$) toward the ray to increase its weight in the rendering equation; conversely, if there is a significant deviation between the two normals, it pushes the Gaussian (e.g., $Position_2$) away from the ray. (b) In contrast, the D-Normal regularizer loss $\mathcal{L}_{dn}$ can move Gaussians either closer to or farther from the GT surface. Here, $Position_1$ and $Position_2$ are 3D positions corresponding to the mean depth of two adjacent pixels (rays), computed via Eq. 5; the D-Normal $\overline{N}_d$ is derived from $Position_1$ and $Position_2$ using Eq. 6. Notably, $\mathcal{L}_{dn}$ relies on the intersection depth, related to Gaussian position **Position** and normal **n**) to encourage $\overline{N}_d$ alignment with the GT normal $N$, ultimately enabling Gaussians to move toward or away from the (GT) surface.

For the plane constraint, any point $o_p$ on plane $(\mathbf{p}, \mathbf{n})$ satisfies $\mathbf{n} \cdot (o_p - \mathbf{p}) = 0$. For the ray representation, a ray originating from the origin is expressed as $o_l = \mathbf{r}t$, where $\mathbf{r}$ denotes the ray direction and $t$ is the distance from the origin. At the intersection $(o_l = o_p)$, solving for the depth along the camera $z$-axis yields:

$$d(\mathbf{n}, \mathbf{p}) = r_z \cdot \frac{\mathbf{n} \cdot \mathbf{p}}{\mathbf{n} \cdot \mathbf{r}}, \tag{18}$$

where $r_z$ is the $z$-component of $\mathbf{r}$.

This $d(\mathbf{n}, \mathbf{p})$ is correlated with both $\mathbf{p}$ and $\mathbf{n}$, ensuring accuracy and enabling D-Normal regularization to backpropagate loss to Gaussian parameters.

## B.2 PROOF PROPOSITIONS

**Proposition 1.1** Supervising the rendered normals cannot effectively influence the positions of Gaussians. The rendered normal $\hat{N}$ is defined as the opacity-weighted average of Gaussian normals. Considering the normal loss $\mathcal{L}_n$, its gradient with respect to the Gaussian position $p_i$ can be expressed via the chain rule as:

$$\frac{\partial \mathcal{L}_n}{\partial p_i} = \frac{\partial \mathcal{L}_n}{\partial \hat{N}} \cdot \frac{\partial \hat{N}}{\partial p_i}. \tag{19}$$

Since each Gaussian normal $n_i$ is determined solely by the rotation parameters, $\frac{\partial n_i}{\partial p_i} = 0$. Thus, the dependency of $\hat{N}$ on $p_i$ originates only from the opacity weights $\alpha_i$:

$$\frac{\partial \hat{N}}{\partial p_i} = \frac{\partial \hat{N}}{\partial \alpha_i} \cdot \frac{\partial \alpha_i}{\partial G(x)} \cdot \frac{\partial G(x)}{\partial p_i}. \tag{20}$$

For a Gaussian distribution

$$G(x) = \exp\left(-\tfrac{1}{2}(x - p_i)^\top \Sigma^{-1}(x - p_i)\right). \tag{21}$$

we obtain

$$\frac{\partial G(x)}{\partial p_i} = -G(x)\,\Sigma^{-1}(x - p_i). \tag{22}$$

In our implementation, following scale regularization, each Gaussian is flattened into an approximate plane, so we approximate $\Sigma^{-1}$ by the identity matrix to emphasize directionality. Hence,

$$\frac{\partial G(x)}{\partial p_i} \approx -G(x)\,(x - p_i). \tag{23}$$

Substituting into Eq.~(16), the resulting position gradient is

$$\frac{\partial \mathcal{L}_n}{\partial p_i} \propto (x - p_i). \tag{24}$$

This indicates that the position update depends only on the spatial offset between the pixel-aligned point $x$ and the Gaussian center $p_i$, without involving the surface normal $n_i$. Consequently, conventional normal supervision can only adjust opacities but fails to drive positions toward the true surface along its normal direction. This explains why rendered-normal supervision alone leads to incomplete geometric optimization.

**Proposition 1.2** Supervising our proposed Depth-Normal (D-Normal) regularizer can effectively influence the positions of Gaussians.

We now consider our proposed D-Normal loss $\mathcal{L}_{dn}$. By definition, the D-Normal $\overline{N}_d$ is computed from the gradients of rendered depth maps. The gradient of $\mathcal{L}_{dn}$ with respect to the Gaussian position $p_i$ follows a three-stage chain rule:

$$\frac{\partial \mathcal{L}_{dn}}{\partial p_i} = \frac{\partial \mathcal{L}_{dn}}{\partial \overline{N}_d} \cdot \frac{\partial \overline{N}_d}{\partial \hat{D}} \cdot \frac{\partial \hat{D}}{\partial p_i}. \tag{25}$$

where $\hat{D}$ denotes the rendered depth.

Since $\hat{D}$ is the opacity-weighted average of Gaussian intersection depths $d_i$, its derivative can be decomposed as:

$$\frac{\partial \hat{D}}{\partial p_i} = \underbrace{\frac{\partial \hat{D}}{\partial \alpha_i} \cdot \frac{\partial \alpha_i}{\partial p_i}}_{\text{(A) Conventional weight term}} + \underbrace{\frac{\partial \hat{D}}{\partial d_i} \cdot \frac{\partial d_i}{\partial p_i}}_{\text{(B) Depth term (new)}}, \tag{26}$$

The depth of a Gaussian–ray intersection is given by:

$$d_i = r_z \cdot \frac{n_i \cdot p_i}{n_i \cdot r}, \tag{27}$$

where $r$ is the viewing ray and $r_z$ its $z$-component. Differentiating with respect to $p_i$ yields:

$$\frac{\partial d_i}{\partial p_i} = r_z \cdot \frac{n_i}{n_i \cdot r}. \tag{28}$$

Thus, the second term (B) in $\partial \hat{D}/\partial p_i$ explicitly involves the surface normal $n_i$. we obtain:

$$\frac{\partial \mathcal{L}_{dn}}{\partial p_i} = \underbrace{\frac{\partial \mathcal{L}_{dn}}{\partial \overline{N}_d} \cdot \frac{\partial \overline{N}_d}{\partial \hat{D}} \cdot \frac{\partial \hat{D}}{\partial \alpha_i} \cdot \frac{\partial G(x)}{\partial G(x)} \cdot \frac{\partial G(x)}{\partial p_i}}_{\text{traditional weight-dependent term}} + \underbrace{\frac{\partial \mathcal{L}_{dn}}{\partial \overline{N}_d} \cdot \frac{\partial \overline{N}_d}{\partial \hat{D}} \cdot \frac{\partial \hat{D}}{\partial d_i} \cdot r_z \cdot \frac{n_i}{n_i \cdot r}}_{\text{new term proportional to } n_i}. \tag{29}$$

The new term proportional to $n_i$ provides a direct mechanism to update the position $p_i$ along the normal direction. As a result, D-Normal supervision not only influences Gaussian rotations (as conventional normal supervision does) but also effectively aligns Gaussian positions with the underlying surface geometry. This theoretical insight explains the substantial geometric improvements observed in our experiments.

**Proposition 2.1**

Supervising our proposed Depth-Consistent D-Normal Regularizer, which incorporates the Pseudo Depth & D-Normal Dual Supervision Mechanism by utilizing both D-Normal maps and pseudo depth maps, can effectively and stably influence Gaussian positions along the normal direction, thereby achieving comprehensive updates of geometric parameters (rotation and position) and significantly improving geometric and reconstruction accuracy.

From Proposition 1.1, conventional rendered normal supervision provides gradients $\frac{\partial \mathcal{L}_n}{\partial p_i} \propto (x - p_i)$, which are independent of Gaussian normals $n_i$ and thus fail to guide positions toward the true surface.

From Proposition 1.2, D-Normal supervision introduces an additional term

$$\frac{\partial \mathcal{L}_{dn}}{\partial p_i} \supset \frac{\partial \mathcal{L}_{dn}}{\partial \overline{N}_d} \cdot \frac{\partial \overline{N}_d}{\partial \hat{D}} \cdot \frac{\partial \hat{D}}{\partial d_i} \cdot r_z \cdot \frac{n_i}{n_i \cdot r}, \tag{30}$$

which is explicitly proportional to the normal $n_i$. This enables position updates along the surface normal direction, thus coupling position and rotation optimization.

However, the reliability of this update depends on the accuracy of rendered depth $\hat{D}$. To further enhance stability, we introduce pseudo depth supervision $\mathcal{L}_{pD}(\hat{D}, D^{\text{pseudo}})$. Its gradient contributes

$$\frac{\partial \mathcal{L}_{pD}}{\partial p_i} = \frac{\partial \mathcal{L}_{pD}}{\partial \hat{D}} \cdot \frac{\partial \hat{D}}{\partial p_i}, \tag{31}$$

which shares the same structural dependence on $\frac{\partial \hat{D}}{\partial p_i}$ as the D-Normal term, and therefore reinforces the normal-dependent component introduced above.

Combining these two complementary signals, the dual supervision mechanism (i) stabilizes depth estimation via pseudo depth, and (ii) ensures normal-consistent position updates via D-Normal. As a result, both rotation and position parameters of Gaussians are comprehensively optimized, yielding improved geometric accuracy in reconstruction.

## C  SUPPLEMENTATION TO THE PARTITIONING STRATEGY

Existing large-scale 3DGS frameworks exhibit two critical limitations: geometric discontinuities at block boundaries introduce visible fusion artifacts, while redundant Gaussians attract non-contributing views that inflate computational loads during block-wise training. Our unified approach addresses both issues through integrated redundancy reduction and geometric continuity enforcement.

The pipeline begins with global pruning of low-impact Gaussians using spatially adaptive scoring (Eq. 15):

$$\mathbf{G}^{\text{pruned}} = G_k \in \mathbf{G} \mid S_k > \theta_{\text{prune}}, \tag{32}$$

This operation targets background and low-contribution primitives that attract irrelevant views. Pre-partition pruning eliminates redundancy propagation to local blocks, significantly reducing computational load.

To enable spatially balanced partitioning, the pruned Gaussians are contracted into a normalized cube $[-1, 1]^3$ using a hybrid contraction function (as in (Wu et al., 2023)), which applies a linear mapping to the foreground region and a nonlinear scaling to the unbounded background. This contraction yields a compact representation of the full scene and facilitates uniform space division.

Within this contracted space, the scene is partitioned into regular blocks. To preserve geometric continuity across adjacent partitions, boundary Gaussians are explicitly duplicated:

$$\mathbf{G}^j_{\text{shared}} = \{G_k \mid \text{dist}(G_k, \partial \mathcal{B}_j) < \delta_{\text{share}}\}. \tag{33}$$

This duplication enforces overlapping geometric constraints near block interfaces, thereby suppressing boundary artifacts during block-wise fusion.

Camera pose assignment for each block $\mathcal{B}_j$ integrates geometric proximity and perceptual contribution through dual evaluation criteria. The geometric criterion assesses physical containment by checking if the contracted camera position $\boldsymbol{p}_{\tau_i}^{\text{ctr}} = \text{contract}(\hat{\boldsymbol{p}}_{\tau_i})$ falls within the block's spatial extent $[\boldsymbol{b}_{j,\min}, \boldsymbol{b}_{j,\max})$, formalized as:

$$B_{\text{geo}}(\tau_i) = \begin{cases} 1 & \boldsymbol{p}_{\tau_i}^{\text{ctr}} \in [\boldsymbol{b}_{j,\min}, \boldsymbol{b}_{j,\max}) \\ 0 & \text{otherwise} \end{cases}, \tag{34}$$

where the contraction operator follows (Liu et al., 2024a).

The perceptual criterion quantifies visual degradation when removing Gaussians $\mathbf{G}_{\mathcal{B}_j}$. By comparing renders $I_{\tau_i}^{\text{full}}$ (full model) and $I_{\tau_i}^{\text{excl-}j}$ (excluding $\mathbf{G}_{\mathcal{B}_j}$) in original space, it computes:

$$B_{\text{vis}}(\tau_i) = \begin{cases} 1 & \text{SSIM}(I_{\tau_i}^{\text{full}}, I_{\tau_i}^{\text{excl-}j}) < 1 - \varepsilon_j \\ 0 & \text{otherwise} \end{cases}, \tag{35}$$

with $\varepsilon_j$ controlling sensitivity to structural loss, identifying perceptually dependent poses.

The final assignment combines both criteria:

$$B(\tau_i) = B_{\text{geo}}(\tau_i) \vee B_{\text{vis}}(\tau_i), \tag{36}$$

ensuring each pose is assigned to blocks it physically occupies or visually relies upon. This establishes efficient view-block correspondence while maintaining rendering consistency.

Table A: Comparison of training times across multiple state-of-the-art methods on the Mill-19 (Yu et al., 2022)and UrbanScene3D (Lin et al., 2022), **Bold** indicates best performance..

| Models | Building Time ↓ | Rubble Time ↓ | Residence Time ↓ | Sci-Art Time ↓ |
|---|---|---|---|---|
| Mega-NeRF | 19:49 | 30:48 | 27:20 | 27:39 |
| Switch-NeRF | 24:46 | 38:30 | 35:11 | 34:34 |
| VastGS † | 03:26 | 02:30 | 03:12 | **03:13** |
| DOGS | 03:51 | 02:25 | 04:33 | 04:23 |
| CityGS-v2 | 04:25 | 03:05 | 04:45 | 04:38 |
| **Ours** | **03:13** | **02:10** | **02:45** | 03:40 |

Table B: Novel View Synthesis Performance Evaluation on the GauU-Scene datasets (Xiong et al., 2024). **Bold** indicates best performance.

| Methods | Residence | | | Russian Building | | | Modern Building | | |
|---|---|---|---|---|---|---|---|---|---|
| | SSIM ↑ | PSNR ↑ | LPIPS ↓ | SSIM ↑ | PSNR ↑ | LPIPS ↓ | SSIM ↑ | PSNR ↑ | LPIPS ↓ |
| NeuS | 0.244 | 15.16 | 0.674 | 0.202 | 13.65 | 0.694 | 0.236 | 14.58 | 0.694 |
| Neuralangelo | NaN | NaN | NaN | 0.328 | 12.48 | 0.698 | NaN | NaN | NaN |
| SuGaR | 0.612 | 21.95 | 0.452 | 0.738 | 23.62 | 0.332 | 0.700 | 24.92 | 0.381 |
| GOF | 0.652 | 20.68 | 0.391 | 0.713 | 21.30 | 0.322 | 0.749 | 25.01 | 0.286 |
| VCR-Gaus | 0.663 | 22.69 | 0.404 | 0.724 | 22.89 | 0.273 | 0.726 | 25.19 | 0.230 |
| 2DGS | 0.703 | 22.24 | 0.306 | 0.788 | 23.77 | 0.189 | 0.776 | 25.77 | 0.202 |
| CityGS-v2 | 0.742 | 23.57 | 0.243 | 0.784 | 24.12 | 0.196 | 0.770 | 25.84 | 0.207 |
| Ours | **0.762** | **23.78** | **0.206** | **0.810** | **24.53** | **0.158** | **0.805** | **26.44** | **0.157** |

# D  MORE EXPERIMENTS

The experimental section of this paper focuses on evaluating the performance of UrbanGS in large-scale scene reconstruction. Through comprehensive comparisons with a variety of baseline methods,

Table C: Detailed geometry evaluation on GauU-Scene datasets (Xiong et al., 2024). "NaN" indicates invalid numerical results, while "FAIL" denotes failure to extract valid mesh. For all metrics, ↑ indicates higher values are better.

| Methods | Campus | | | Village | | | College | | |
|---|---|---|---|---|---|---|---|---|---|
| | P ↑ | R ↑ | F1 ↑ | P ↑ | R ↑ | F1 ↑ | P ↑ | R ↑ | F1 ↑ |
| NeuS | FAIL | FAIL | FAIL | FAIL | FAIL | FAIL | FAIL | FAIL | FAIL |
| Neuralangelo | NaN | NaN | NaN | NaN | NaN | NaN | NaN | NaN | NaN |
| SuGaR | 0.321 | 0.272 | 0.294 | 0.354 | 0.253 | 0.295 | 0.409 | 0.271 | 0.326 |
| VCR-Gaus | 0.478 | 0.312 | 0.379 | 0.492 | 0.412 | 0.448 | 0.456 | 0.361 | 0.401 |
| 2DGS | 0.389 | 0.304 | 0.341 | 0.442 | 0.283 | 0.345 | 0.340 | 0.182 | 0.237 |
| GOF | FAIL | FAIL | FAIL | FAIL | FAIL | FAIL | FAIL | FAIL | FAIL |
| PGSR | 0.464 | 0.355 | 0.403 | 0.535 | 0.445 | 0.486 | 0.349 | 0.349 | 0.354 |
| CityGS-X | 0.505 | 0.361 | 0.421 | 0.545 | 0.443 | 0.489 | 0.559 | 0.371 | 0.446 |
| CityGaussianV2 | 0.486 | 0.383 | 0.428 | 0.580 | 0.503 | 0.543 | 0.577 | 0.373 | 0.453 |
| Ours | 0.492 | 0.388 | 0.435 | 0.567 | 0.512 | 0.538 | 0.564 | 0.381 | 0.456 |

Table D: Ablation study of different priors on the Modern Building dataset.

| Priors | | | | Rendering Quality | | | Geometric Quality | | |
|---|---|---|---|---|---|---|---|---|---|
| Dav2 | MiDaS | Dsine | GeoWizard | SSIM ↑ | PSNR ↑ | LPIPS ↓ | P ↑ | R ↑ | F1 ↑ |
| ✓ | | ✓ | | **0.805** | 26.44 | **0.157** | 0.663 | 0.404 | **0.503** |
| ✓ | | | ✓ | 0.802 | **26.48** | 0.163 | **0.665** | 0.399 | 0.498 |
| | ✓ | ✓ | | 0.798 | 26.33 | 0.161 | 0.645 | **0.410** | 0.501 |
| | ✓ | | ✓ | 0.785 | 26.12 | 0.166 | 0.658 | 0.392 | 0.491 |

the effectiveness of UrbanGS is validated across three key aspects: training efficiency, novel view synthesis quality, and geometric accuracy.

## D.1 TRAINING EFFICIENCY

As shown in table A,in training time comparison experiments conducted on diverse scenes such as Building, Rubble, Residence, and Sci-Art, UrbanGS demonstrates significant efficiency gains. For instance, in the Building scene (Yu et al., 2022), UrbanGS completes training in only 3 hours and 13 minutes, substantially outperforming Mega-NeRF (19 hours 49 minutes) (Turki et al., 2022) and Switch-NeRF (24 hours 46 minutes) (Mi & Xu, 2023). Even when compared with more efficient baselines such as VastGS† (Lin et al., 2024), UrbanGS consistently achieves competitive or superior training times across most scenes.

Table E: Effect of block partitioning on the *Russian* dataset. Memory (Mem) in GB, Time in minutes. **Bold** indicates best performance.

| Block/GPU | PSNR↑ | SSIM↑ | LPIPS↓ | F1↑ | Mem↓ | Time↓ |
|---|---|---|---|---|---|---|
| 2/2 | 23.43 | 0.779 | 0.215 | 0.518 | 25.2 | 170 |
| 4/4 | 24.55 | 0.804 | 0.201 | 0.539 | 20.1 | 140 |
| 8/8 | **24.66** | **0.813** | **0.184** | **0.546** | **14.4** | **122** |

## D.2 NOVEL VIEW SYNTHEIS

As shown in table B, the novel view synthesis performance is also evaluated on the GauU-Scene dataset (Xiong et al., 2024), UrbanGS outperforms competing methods in all tested scenes using SSIM, PSNR (higher is better), and LPIPS (lower is better) as evaluation metrics. Specifically, in the Residence scene, it achieves SSIM 0.762, PSNR 23.78, and LPIPS 0.206; in the Russian

Building scene, SSIM 0.810, PSNR 24.53, and LPIPS 0.158; and in the Modern Building scene, SSIM 0.805, PSNR 26.44, and LPIPS 0.157. These results consistently surpass baselines such as SuGaR (Guédon & Lepetit, 2024) and GOF (Yu et al., 2024b).

### D.3 GEOMETRIC ACCURACY

To further assess the generalization of our method to large-scale urban scenes, we additionally evaluate geometry quality on the GauU-Scene dataset (Xiong et al., 2024). As summarized in Table C, we report precision (P), recall (R), and F1 score across three subsets (Campus, Village, and College). Several NeRF- and 3DGS-based baselines either produce invalid numerical results ("NaN") or fail to extract a valid mesh ("FAIL"), highlighting the difficulty of this benchmark. In contrast, our method consistently reconstructs valid meshes and achieves the best or highly competitive performance on all metrics. As a qualitative complement to the numerical results, Figure B and Figure C compares the reconstructed meshes of representative methods on multiple GauU-Scene subsets. Baseline approaches often suffer from over-smoothed surfaces, broken structures, or missing fine-scale details, particularly around building facades and road layouts, whereas our method produces more complete and coherent geometry with sharper boundaries.

### D.4 MORE ABLATIONS

As shown in Figure D, we conducted ablation studies on the Campus dataset. When the Depth-Consistent D-Normal Regularization module and Partitioning Strategy module are ablated, significant differences are observed in both rendered normal maps and rendered depth maps compared with our full method, demonstrating the substantial effectiveness of these modules in the proposed approach. To more clearly demonstrate the effectiveness of our module in pushing 3D points along the normal direction, we conduct an experiment on a small scene. As shown in Figure E, when the Depth-Consistent D-Normal Regularization module is removed, the point cloud on object surfaces becomes highly scattered. With our regularizer enabled, the points are driven toward the underlying surfaces, resulting in a much more compact point cloud and a significantly cleaner reconstruction.

Table D studies the influence of using different depth estimators (Dav2 (Hu et al., 2024), MiDaS (Ranftl et al., 2020)) and the normal prior (GeoWizard (Fu et al., 2024), Dsine (Bae & Davison, 2024)) on the Modern Building dataset. Across all combinations, the rendering metrics remain very close (SSIM $\approx$ 0.79–0.81, PSNR $\approx$ 26.1–26.5, LPIPS $\approx$ 0.157–0.166) and the geometric quality (P/R/F1) only fluctuates within a small range. This indicates that our framework is not sensitive to the particular choice of depth or normal prior. Thanks to the explicit depth-consistency and normal-consistency constraints, the optimization can effectively correct the bias of different priors and consistently recover high-quality geometry.

As shown in Table E, under the experimental scenario of the Russian dataset, this table presents the impacts of different "Block/GPU count" configurations on model performance, memory consumption, and training time. As the configuration is scaled up from 2/2 (2 blocks, 2 GPUs) to 8/8 (8 blocks, 8 GPUs), the model performance is gradually optimized: PSNR increases from 23.43 to 24.66, SSIM rises from 0.779 to 0.813, LPIPS decreases from 0.215 to 0.184, and F1 improves from 0.518 to 0.546. Meanwhile, resource consumption is significantly reduced, with memory usage dropping from 25.2 GB to 14.4 GB and training time shortening from 170 minutes to 122 minutes. This demonstrates the positive role of the parallel training strategy—where the number of blocks matches the number of GPUs—in balancing "performance-efficiency".

Table G reports the effect of removing individual components in our block partition strategy on the Russian scene of the GauU-Scene dataset (Xiong et al., 2024). Starting from our full model, discarding the global pruning term in Eq. 32 leads to a clear increase in the number of Gaussians (2.45M → 3.01M), longer training time and higher memory usage, together with a slight drop in both rendering and geometric quality. This confirms that the spatially adaptive pruning not only reduces redundancy but also facilitates optimization. Removing the boundary-duplication rule in Eq. 33 also degrades SSIM, LPIPS, and F1, indicating that sharing Gaussians across neighboring blocks is important for suppressing block-boundary artifacts. When the geometric pose-assignment criterion in Eq. 34 is disabled, the performance drops most significantly (e.g., PSNR and F1 both

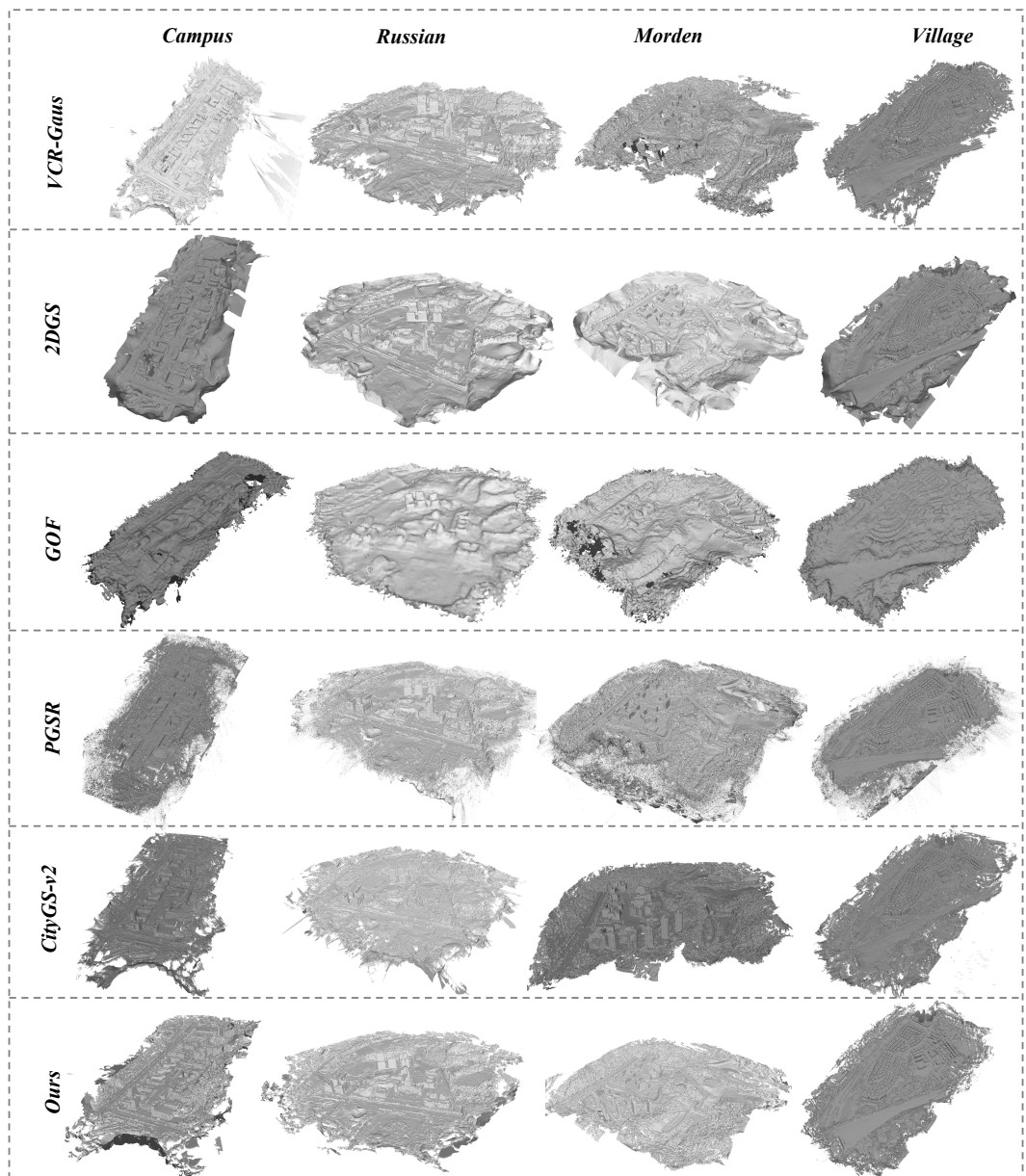

Figure B: Visual comparison of meshes from state-of-the-art (SOTA) methods.

decrease considerably) while the computational cost increases, showing that aligning camera frusta with physically relevant blocks is crucial for both fidelity and efficiency. Finally, removing the perceptual criterion in Eq. 35 causes a moderate decline in rendering quality and F1, demonstrating that perceptual filtering helps retain poses that are visually important for each block. Taken together, these results show that all components of our partition strategy contribute to the overall trade-off, and the full design achieves the best balance between reconstruction quality and resource consumption.

As shown in Figure G, on the Sci-Art scenes (Lin et al., 2022) we observe that 3DGS-based methods with explicit geometry optimization often yield lower rendering quality than the original 3DGS. These scenes contain many aerial-style images dominated by distant sky regions with weak or ambiguous geometry. In such backgrounds, geometry-optimized variants tend to degrade the sky appearance, producing coarse color blotches and unnatural boundaries in the rendered views. In contrast, although the original 3DGS is also imperfect in sky modeling, its results still vaguely preserve

|  | *Campus* | *Village* | *Morden* |
|---|---|---|---|
| *2DGS* | | | |
| *GoF* | | | |
| *VCR-GauS* | | | |
| *PGSR* | | | |
| *CityGS-x* | | | |
| *CityGS-v2* | | | |
| *Ours* | | | |

Figure C: Qualitative mesh and texture comparison between SOTA and our method on the Campus, Village, and Morden Buliding scenes (Xiong et al., 2024).

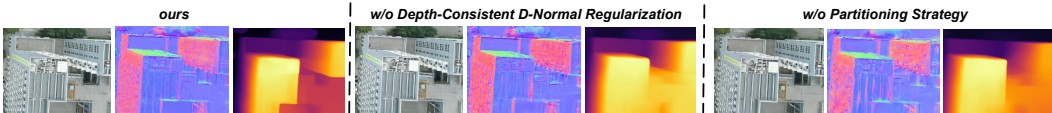

Figure D: Ablation experiments on the Campus Dataset Lin et al. (2022)

cloud layers and building silhouettes. This discrepancy highlights a limitation of current geometry optimization objectives when applied to background regions lacking clear geometric structure.

Table H presents the ablation study results of the two key hyperparameters $\gamma_d$ and $\tau$ in the geometry-aware confidence mechanism, conducted on the Modern Building (Lin et al., 2022). scene. The baseline configuration ($\gamma_d = 0.1$, $\tau = 0.01$) achieves the optimal performance across all evaluation

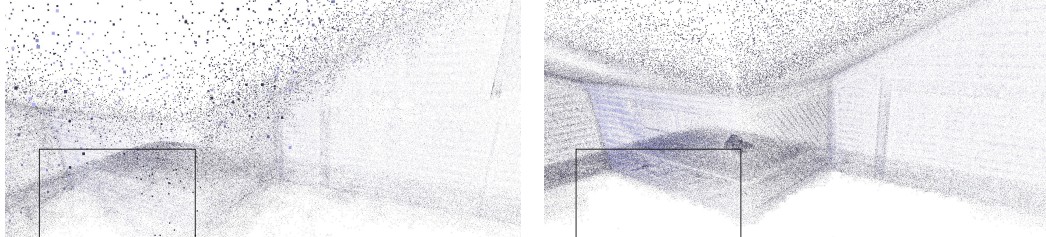

Figure E: Qualitative ablation for the Depth-Consistent D-Normal Regularizer. We visualized the centers of Gaussian ellipsoids in a 3D scene. In the left figure, the Depth-Consistent D-Normal Regularizer is disabled, while the right figure demonstrates the results with our proposed regularization. In comparison, the left figure exhibits a notable number of Gaussian ellipsoids floating off the surface. Our proposed Depth-Consistent D-Normal Regularizer effectively pushes the 3D Gaussians toward the surface, resulting in a cleaner reconstruction.

Table F: Quantitative Analysis of Weight Configuration Ablation Study

| Weight Configuration | PSNR ↑ | SSIM ↑ | LPIPS ↓ | F1 ↑ |
|---|---|---|---|---|
| (1.2, 1.0, 0.8) | **26.44** | **0.805** | **0.157** | **0.503** |
| (1.0, 1.0, 1.0) | 26.19 | 0.791 | 0.172 | 0.487 |
| (0.0, 1.0, 1.0) | 24.32 | 0.763 | 0.215 | 0.432 |
| (1.0, 0.0, 1.0) | 25.12 | 0.778 | 0.198 | 0.468 |
| (1.0, 1.0, 0.0) | 25.87 | 0.793 | 0.169 | 0.485 |
| (1.2, 1.0, 1.0) | 25.95 | 0.782 | 0.185 | 0.451 |

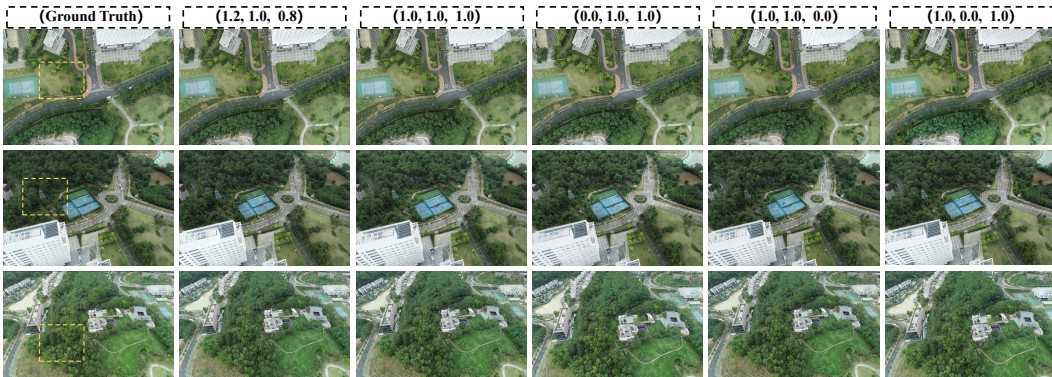

Figure F: Ablation experiments on the Morden Building Dataset Xiong et al. (2024)

metrics, with PSNR of 26.44 and F1-score of 0.503. Decreasing $\gamma_d$ to 0.05 alone leads to noticeable performance degradation (PSNR drops by 0.32, F1-score drops by 0.018), indicating that excessive sensitivity to depth gradient consistency suppresses valid geometric signals. Similarly, increasing $\tau$ to 0.02 alone also causes performance deterioration (PSNR drops by 0.16, F1-score drops by 0.011), suggesting insufficient suppression of depth errors adversely affects reconstruction quality. The worst performance occurs when both parameters are modified ($\gamma_d = 0.15$, $\tau = 0.005$), with PSNR and F1-score decreasing by 0.55 and 0.025 respectively, validating the coupling relationship between the two hyperparameters and the rationality of the baseline selection. These results comprehensively demonstrate that our chosen hyperparameter combination achieves the optimal balance between geometric consistency and error suppression.

## D.5 LINEAR WEIGHTED PRUNING

We refer to the conventional weighting scheme as *Linear Weighted Pruning (LWP)*, where the importance score is computed as a linear combination of the three attributes: $S_i^{\text{LWP}} = \alpha \cdot \phi_i + \beta \cdot \tau_i + \gamma \cdot w_{v,i}$. To analyze the impact of the weighting hyperparameters, we conduct an ablation study on the LWP formulation. As summarized in Table F, we systematically vary $\alpha$, $\beta$, and $\gamma$ and evaluate the re-

Table G: Ablation Results of Block Partition Strategy on Russian Scene Dataset (Xiong et al., 2024).**Bold** indicates best performance.

| Method | Rendering Quality | | | Geometric Quality | | | Training Statistics | | | |
|---|---|---|---|---|---|---|---|---|---|---|
| | PSNR↑ | SSIM↑ | LPIPS↓ | P↑ | R↑ | F1↑ | GS (millions)↓ | Time (min)↓ | Size (MB)↓ | Mem (GB)↓ |
| **Effect of Removing Individual Components** | | | | | | | | | | |
| **baseline (ours)** | **24.66** | **0.813** | **0.184** | **0.568** | **0.525** | **0.546** | **2.45** | **122** | **314.24** | **14.4** |
| baseline w/o Eq. 32 | 24.43 | 0.797 | 0.201 | 0.562 | 0.518 | 0.539 | 3.01 | 142 | 429.41 | 17.5 |
| baseline w/o Eq. 33 | 24.51 | 0.802 | 0.198 | 0.564 | 0.513 | 0.537 | 2.44 | 129 | 314.24 | 15.1 |
| baseline w/o Eq. 34 | 22.32 | 0.764 | 0.231 | 0.531 | 0.498 | 0.513 | 2.56 | 157 | 334.31 | 20.3 |
| baseline w/o Eq. 35 | 24.42 | 0.808 | 0.188 | 0.566 | 0.521 | 0.543 | 2.46 | 125 | 314.31 | 14.7 |

Table H: Ablation Study of Geometry-Aware Confidence Hyperparameters

| Hyperparameter Settings | PSNR ↑ | F1-score ↑ |
|---|---|---|
| Baseline ($\gamma_d = 0.1$, $\tau = 0.01$) | 26.44 | 0.503 |
| Only $\gamma_d = 0.05$ | 26.12 | 0.485 |
| Only $\tau = 0.02$ | 26.28 | 0.492 |
| Both modified ($\gamma_d = 0.15$, $\tau = 0.005$) | 25.89 | 0.478 |

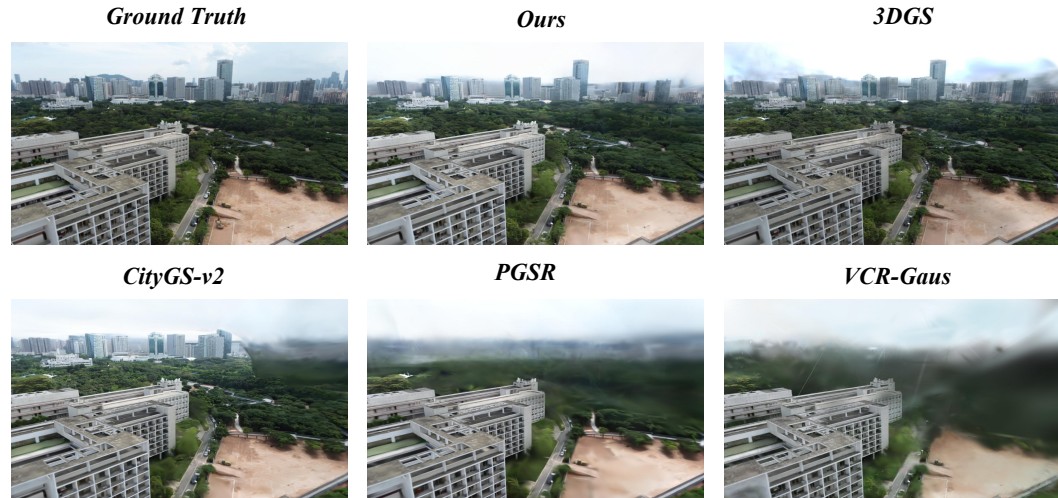

Figure G: Qualitative mesh and texture comparison between SOTA and our method on Art-Sci Scene (Lin et al., 2022).

sulting rendering quality (PSNR, SSIM, LPIPS) and geometric accuracy (F1-score). The optimal configuration ($\alpha = 1.2$, $\beta = 1.0$, $\gamma = 0.8$) achieves the best performance across all metrics: PSNR 26.44, SSIM 0.805, LPIPS 0.157, and F1-score 0.503. Compared to an equal-weight baseline ($\alpha = \beta = \gamma = 1.0$), this setting improves PSNR by 0.55 dB and F1-score by 0.016.

Univariate ablation reveals the distinct roles of each term: setting $\alpha = 0.0$ (removing the ray-intersection frequency) causes a 14.1% drop in F1-score, underscoring its importance for multi-view consistency; setting $\beta = 0.0$ (ignoring opacity) increases LPIPS by 26.1%, indicating a direct impact on visual quality; and using $\gamma = 1.0$ (i.e., a linear volume weight instead of the sub-linear one) degrades all metrics, confirming that aggressive volume weighting introduces excessive redundancy. Thus, the chosen weights strike a balance between rendering fidelity, geometric completeness, and computational efficiency.

Qualitative results in Figure F show that rendered views remain visually consistent across different weight combinations, with no catastrophic failures even for suboptimal settings. This demonstrates that the LWP scheme is reasonably robust to moderate variations in $\alpha, \beta, \gamma$, provided they stay

within a sensible range. Nevertheless, the need for manual hyperparameter tuning in LWP motivates our proposed multiplicative scoring (Eq. 15), which eliminates such tuning while preserving or improving performance.

## E    LIMITATIONS

Although UrbanGS demonstrates advantages in large-scale reconstruction, it still exhibits certain limitations. Its geometric regularization relies on monocular depth/normal priors derived from pre-trained networks, which may propagate estimation errors into the reconstruction—particularly in regions with weak textures or extreme lighting conditions. Additionally, the method primarily focuses on static environments and does not explicitly model dynamic objects commonly found in urban scenes. Future work will aim to mitigate dependency on monocular priors through multi-view geometric consensus and extend the framework to dynamic urban objects via explicit motion modeling.

## F    USE OF LARGE LANGUAGE MODELS

A large language model (LLM) was used solely for language-level assistance, such as improving readability, fluency of the text and formatting LaTeX tables and retrieve related works. The research ideas, experiments, and results are entirely the work of the authors, who bear full responsibility for the content of this submission.

