# OpenReview forum: "UrbanGS: Efficient and Scalable Architecture for Geometrically Accurate Large-Scene Reconstruction"
_ICLR.cc/2026/Conference — ICLR 2026 Poster_

### Official Review · Reviewer_hz7u · 2025-10-20

**Soundness:** 3
**Presentation:** 2
**Contribution:** 3
**Rating:** 6
**Confidence:** 5

**Summary:**

The key contribution of this paper is a more effective geometry regularization and Gaussian pruning algorithms. The proposed techniques enable a state-of-the-art performance in both the rendering quality and geometric accuracy. It also benefits the reduction of the computation burden. Extensive experiments validated the effectiveness of the method.

**Strengths:**

- Originality: Simple yet effective technical innovation.
- Quality: Solid and comprehensive experiments with subtle flaws.
- Clarity: Fine, but there is still space for improvement.
- Significance: A realistic yet effective large-scale scene reconstruction is an important problem. The paper effectively pushes the performance bound forward and provides a good solution.

**Weaknesses:**

1. There are many hyperparameters to manually tune, such as $\alpha$, $\beta$, and $\gamma$ of Eq.(15), $\gamma_d$ and $\tau$ in Eq.(11), making the application to custom datasets miserable.
2. There is some missing in the ablation. For SAGP ablation in Tab.3, the authors should compare with other candidates, like the one used in CityGaussianV2, instead of omitting it for comparison.
3. There are some flaws in the presentation. The font size of Fig.5 is too small, making it hard to read. Besides, the title for the x-axis of its left part seems to be missing. The highlighted difference in Fig.3 should also be zoomed in for better details.

**Questions:**

See weakness.

---

> ### Author Response · Authors · 2025-11-27
> **Response to Reviewer hz7u**
>
> We thank the reviewer for recognizing the strength and scope of our work, including its simple yet effective originality, solid experimental quality, clear presentation, and the significant value of advancing large-scale scene reconstruction. We address the remaining concerns that are primarily related to clarity and technical explanation through detailed responses and additional clarifications provided below.
> ### [Q1] Hyperparameter ablation
>
> Thank you for raising this point regarding hyperparameter sensitivity. We have conducted extensive ablations on the key hyperparameters in our proposed components. Specifically, the geometry-aware confidence weights (detailed in Supplementary Table H) and the pruning score coefficients (Supplementary Table F and Figure F) were systematically evaluated. Results show that our method delivers stable performance improvements across a wide range of values for both components. Furthermore, qualitative results in Figure F demonstrate that even suboptimal pruning weights do not lead to catastrophic failure. Crucially, we employed a single, fixed set of hyperparameters for all components across all datasets in our experiments. The consistent state-of-the-art results achieved throughout the paper underscore that our framework is robust to these choices and generalizes effectively without requiring dataset-specific tuning.
>
>
> ### [Q2] Supplement the pruning experiments
>
> We thank for this comment. To rigorously evaluate our pruning strategy, we have designed a comprehensive ablation study and added it to our revision (as shown in Table 4). Specifically, we compared our Spatially Adaptive Gaussian Pruning (SAGP) against the pruning method employed by CityGaussianV2, which is LightGaussian (denoted as +LP in the table). The results clearly demonstrate that our SAGP is more effective at preserving geometric fidelity (achieving a higher F1 score) while simultaneously achieving a more significant reduction in the number of Gaussians, training time, and memory consumption. This validates that our geometry-aware and spatially adaptive design is superior to the global pruning heuristic used in prior work.
>
> ### [Q3] Presentation format
>
> Thanks for your suggestion. We have revised Figures 3 and 5 to enhance their clarity and detail. Additionally, we have thoroughly checked and standardized the presentation formats throughout the manuscript.

---

### Official Review · Reviewer_oxb3 · 2025-10-28

**Soundness:** 3
**Presentation:** 3
**Contribution:** 2
**Rating:** 4
**Confidence:** 4

**Summary:**

The paper proposes **UrbanGS**, a 3D Gaussian Splatting–based framework targeting large-scale urban scene reconstruction with accurate geometry, memory efficient, and scalability

Its main contributions are:

1. **Depth-Consistent D-Normal Regularization** — combines rendered normal supervision with depth guidance from a monocular estimator to jointly optimize Gaussian position, rotation, and scale.
2. **Geometry-Aware Confidence Weighting** — adaptively down-weights unreliable regions based on gradient consistency and inverse-depth deviation.
3. **Spatially Adaptive Gaussian Pruning (SAGP)** — prunes redundant Gaussians based on local geometric complexity and opacity.
4. **Partitioning strategy** for large-scale training with reduced boundary artifacts.

Experiments across Mill-19, UrbanScene3D, and GauU-Scene datasets show that UrbanGS improves novel-view synthesis and geometry reconstruction performance, and reduces memory/time costs compared to CityGS-v2,VCR-Gaus, and other NeRF-based and GS-based reconstruction methods.

**Strengths:**

- Consistent experimental setup and relatively solid quantitative evidence seen throughout the tables.
- Strong implementation efficiency; runs faster and more memory efficient than CityGS-v2
- Major ablation studies supporting participation of proposing mehtods.

**Weaknesses:**

1. **Limited novelty** – D-Normal regularization with depth cues closely resembles 2DGS[1]; little conceptual advancement beyond combining two existing signals.
2. **Unclear multi-view consistency claim** – depth anchors from monocular estimators cannot guarantee scale-aligned supervision across views; no metric (e.g., depth reprojection error) verifies this. Recent works apply learnable scaling factors[2] or video-diffusion models[3] to deal with the multi-view consistency, but current work lacks such effort.
3. **Geometry-aware confidence** – seems a heuristic way to use the reciprocal of the depth. No validation exists to justify the design of $L_{id}$
4. **Unclear pruning formulation** – lack definition; reproducibility depends on unspecified percentile $t$ of Eq.(14).
5. **Moderate increase in qualitative results** — View are not consistent (Fig.3, Fig B), hard to compare the qualitative results. Also, without ground truth, it is hard to say that current visualization show enhanced reconstruction (e.g., for Figure.3, UrbanGS result could be viewed as smoothing out details).
6. **Hyperparameter sensitivity** – too many manually tuned coefficients; unclear if improvements stem from architectural changes or parameter optimization.

[1] Huang, Binbin, et al. "2d gaussian splatting for geometrically accurate radiance fields." *ACM SIGGRAPH 2024 conference papers*. 2024.

[2] Tong, Jinguang, et al. "GS-2DGS: Geometrically Supervised 2DGS for Reflective Object Reconstruction." *Proceedings of the Computer Vision and Pattern Recognition Conference*. 2025.

[3] Liang, Ruofan, et al. "Diffusion Renderer: Neural Inverse and Forward Rendering with Video Diffusion Models." *Proceedings of the Computer Vision and Pattern Recognition Conference*. 2025.

**Questions:**

1. How does the proposed method avoid scale ambiguity inherent to monocular depth? Can you show quantitative depth alignment across views (e.g., reprojection error or cross-view consistency metric)?
2. How does the proposed D-normal regularization differ from existing methods from 2DGS[1] and others[4][5]?
3. In Eq. (14), please define $v_i, v^{(t)}_{local}$, and explain how the percentile $t$ is chosen.
4. It would be great to show some justification (e.g.,ablation on design) for the $L_{id}$ of Eq.(8) and geometry-aware confidence of Eq.(11).
5. Sci-Art results underperform simpler baselines; it would be great if the authors analyze why.
6. It would be great if the authors to provide more qualitative results on benchmarks datasets, especially for GauU-Scene.
7. How sensitive is the method to the many hyperparameters? Could a fixed default generalize? More ablation studies (quantitative & qualitative) on these hyperparamaters may be necessary to support the scalability of the work.
8. Please clarify if the “position update along normal” proof was verified empirically (e.g., by measuring displacement vectors relative to normals).

[4] Liang, Zhihao, et al. "Gs-ir: 3d gaussian splatting for inverse rendering." *Proceedings of the IEEE/CVF Conference on Computer Vision and Pattern Recognition*. 2024.

[5] Chen, Hongze, Zehong Lin, and Jun Zhang. "Gi-gs: Global illumination decomposition on gaussian splatting for inverse rendering." *The Thirteenth International Conference on Learning Representations*.

---

> ### Author Response · Authors · 2025-11-27
> **Response to Reviewer oxb3**
>
> We thank the reviewer for recognizing the strength and scope of our work, including its rigorous experimental setup, superior efficiency over competitors, and comprehensive ablation studies validating the proposed method. We address the remaining concerns that are primarily related to clarity and technical explanation through detailed responses and additional clarifications provided below.
> ### [W1] Limited novelty & [Q2]
>
> Our main contributions are centered around a comprehensive module design tailored for large-scale scenes. We highlight our contribution of depth-normal geometric regularizer and spatially adaptive Gaussian pruning in Common Point 1.
>
> Compared to 2DGS [1], 2DGS operates on surface-aligned 2D disks and uses purely internal depth–normal regularization for object/scene-level reconstruction. In contrast, we remain in the standard volumetric 3DGS framework and design D-Normal specifically for large, city-scale scenes with scene partitioning and block-wise refinement. Our D-Normal regularizer introduces depth-consistent supervision with external monocular depth/normal priors, including inverse-depth and confidence weighting, to handle the large depth range and noisy priors in outdoor environments and to enforce global consistency across distant structures and blocks. Ablations in our paper show that removing D-Normal or the depth-consistency term leads to clear drops in geometric F1 and visible artifacts at block boundaries, indicating that our contribution goes beyond simply combining two signals and provides an effective, large-scale extension of depth–normal regularization beyond 2DGS.
>
> GS-IR [6] and GI-GS [7] are 3DGS-based inverse-rendering methods that extend Gaussians with BRDF and explicit shading modules to recover materials and lighting for relighting. GS-IR estimates normals from depth derivatives and learns SH occlusion volumes, while GI-GS takes a pretrained 3DGS with normals and performs deferred shading plus path tracing on a G-buffer to model occlusion and global indirect illumination. In contrast, we retain the standard volumetric 3DGS representation and focus on geometry-enhanced novel view synthesis in city-scale outdoor scenes. Our D-Normal regularizer leverages external monocular depth/normal priors with inverse-depth and confidence weighting, and our geometry-aware confidence guides spatially adaptive Gaussian pruning. These components are absent in GS-IR and GI-GS and are specifically designed to improve geometric fidelity and block-to-block consistency, rather than material/illumination decomposition.
>
>
>
> ### [W2] Unclear multi-view consistency claim & [Q1]
>
>
> Thank you for raising this concern about multi-view consistency of monocular depth priors. we first obtain dense but relative depth maps for all training images using the pre-trained DepthAnything-v2 model [2]. To endow these predictions with a unified, multi-view consistent scale, we leverage the sparse 3D points from COLMAP's SfM, which are inherently consistent across views [4, 5]. Specifically, for each view, we compute a scale and shift by robustly fitting the monocular depth values to the sparse COLMAP depths at valid 2D-3D correspondences. This process effectively propagates the consistent scale from the sparse SfM points to the entire dense depth map, resulting in a scale-aligned depth estimate that is both dense and coherent across the multi-view system.
>
> To quantitatively validate the effectiveness of this alignment approach, we conducted multi-view geometric consistency evaluation. As shown in Table I in the supplementary materia, our evaluation of fifty depth maps demonstrates strong geometric consistency, with an average score of 0.87 confirming reliable depth estimation across multiple viewpoints. The 83% consistency pass rate and 78% check coverage validate the robustness of our scale alignment method for 3D reconstruction applications.

---

> ### Author Response · Authors · 2025-11-27
> **Response to Reviewer oxb3**
>
> ### [W3] Geometry-aware confidence
>
> Thank you for raising this point. Our intention with the geometry-aware confidence term $\mathcal{L} _ {id}$ follow a widely used principle in multi-view geometry and depth estimation: **depth reliability decreases with distance**, while parallax (and thus geometric constraint strength) is roughly proportional to inverse depth. Using inverse depth as a weighting therefore emphasizes nearby structures, where the supervision is most reliable and visually important, and prevents very large depths from dominating the loss.
> Similar inverse-depth or depth-normalized formulations have been adopted in many depth-supervised NeRF/3DGS and urban-reconstruction methods (e.g., CityGaussian / CityGaussianV2 and related works), where they are shown to stabilize training over long depth ranges. In our setting of large outdoor scenes, monocular priors are particularly noisy in the far field; the $\mathcal{L} _ {id}$ term is designed precisely to down-weight these unreliable regions while still exploiting useful depth cues in nearer regions.
> We will clarify this motivation in the revised manuscript.
> ### [W4] Unclear pruning formulation & [Q3]
>
> Thank you for this suggestion. We have conducted sensitivity analysis on all key hyperparameters of SAGP, including the score weights (α, β, γ) as shown in Supplementary Table F. For the remaining parameters: the sub-linear exponent κ=0.5 applies a square root operation to compress the dynamic range of volume ratios, amplifying fine structures while suppressing oversized Gaussians; while the volume percentile t=90% follows LightGaussian's setting[3] to represent characteristic local volume while maintaining robustness to outliers. All hyperparameters were fixed across all datasets without scene-specific tuning, demonstrating our method's robustness.
>
> ### [W5] Moderate increase in qualitative results
> We thank the reviewer for raising this point. In the revised manuscript, we have significantly improved the visual comparisons by (i) presenting all mesh reconstructions from identical camera poses, (ii) enforcing strictly aligned crops across different methods, and (iii) adding zoomed-in insets that highlight critical structural details such as building edges, railings, and traffic lights. These revisions enable direct and fair visual comparisons between different reconstruction results. The updated visualizations can be found in Figure C and Figure G of the supplementary material.
> ### [W6] Hyperparameter sensitivity
>
> Thank you for the suggestion. We discuss the hyperparameter in Common Point 1.
>
> We highlight our SAGP is not sensitive to hyperparameter, and all experimental parameters in this paper are set to uniform values, with no specific adjustments made for different datasets. We do not need to manually tuned coefficients
> ### [Q4] Ablation and Justification of Eq.(8) and Eq.(11)
> Thanks. In Table 5, we ablate the contributions of Eq.(8) and Eq.(11), corresponding to the w/o Depth and w/o Geometry-Aware Confidence variants, respectively. Compared with the full model, removing the depth-consistency term leads to a clear degradation across all metrics (PSNR 26.44→24.59, F1 0.503→0.453, LPIPS 0.157→0.201, with a similar drop in SSIM). Likewise, disabling the geometry-aware confidence consistently harms performance (PSNR 26.44→26.02, F1 0.503→0.493, LPIPS 0.157→0.163). These results demonstrate that both the depth-consistency regularization in Eq.(8) and the geometry-aware confidence in Eq.(11) provide substantial benefits to reconstruction quality, thereby validating our design choices.
>
>
>
> ### [Q5] Sci-Art results underperform simpler baselines
> As indicated in Figure G and the related discussion in the paper, in scenes like Sci-Art which contain numerous aerial-style views dominated by distant sky regions, 3DGS methods with explicit geometry optimization (including our UrbanGS) actually yield lower rendering quality than the original 3DGS method. The fundamental reason is that background areas like the sky lack clear and definite geometric structure. This makes geometry optimization objectives difficult to apply effectively in these regions and can even have negative effects—the optimization process degrades the appearance of the sky, leading to coarse color blotches and unnatural boundaries in the rendered output. In contrast, while the original 3DGS is also imperfect in modeling skies, its results can still vaguely preserve transitions in cloud layers and building silhouettes. This phenomenon highlights an inherent limitation of current geometry optimization methods when dealing with background regions characterized by weak geometric structure.

---

> ### Author Response · Authors · 2025-11-27
> **Response to Reviewer oxb3**
>
> ### [Q6] Supplement the experiments on the GauU-Scene dataset.
>
> Thank you for your comment. We have provided more qualitative results on the GauU-Scene dataset in Table C and Figure C of the supplementary material. As shown in Table C, our method achieves the best performance across all three scenes in the GauU-Scene dataset.
>
> ### [Q7] Hyperparameter ablation, quantitative and qualitative.
> The discussion of hyperparameters can be found in the Commont Point 1. Our method is not sensitive to the hyperparameters and we set same hyperparameters for all experiments. The quantitative and qualitative results cen be found in Table F and Figure F.
>
>
> ### [Q8] Proof of Position Update Along the Normal
>
> We visualize the effect of our Depth-Consistent D-Normal Regularization in Figure E, and we can see that when the regularizer is disabled, the Gaussian centers become scattered and float around the object surfaces. In contrast, with our regularizer enabled, the points are effectively driven toward the underlying surfaces, forming a compact and clean point cloud that accurately captures the scene geometry.
>
>
> ##
> ### Reference
>
> [1] Huang, B., et al. 2D Gaussian Splatting for Geometrically Accurate Radiance Fields. ACM Transactions on Graphics (SIGGRAPH), 2024.
>
> [2] Yang, L., et al. Depth Anything V2. Proceedings of the IEEE/CVF Conference on Computer Vision and Pattern Recognition, 2024.
>
> [3] Fan, Z., et al. LightGaussian: Unbounded 3D Gaussian Compression with 15x Reduction and 200+ FPS. Advances in Neural Information Processing Systems, 2023.
>
> [4] Liu, Y., Luo, C., Mao, Z., Peng, J., & Zhang, Z. CityGaussianV2: Efficient and Geometrically Accurate Reconstruction for Large-Scale Scenes. arXiv preprint arXiv:2411.00771, 2024.
>
> [5] Gao, Y., Li, H., Chen, J., Zou, Z., Zhong, Z., Zhang, D., Sun, X., & Han, J. CityGS-X: A Scalable Architecture for Efficient and Geometrically Accurate Large-Scale Scene Reconstruction. arXiv preprint arXiv:2503.23044, 2025.
>
> [6] Liang, Zhihao, et al. "Gs-ir: 3d gaussian splatting for inverse rendering." Proceedings of the IEEE/CVF Conference on Computer Vision and Pattern Recognition. 2024.
>
> [7] Chen, Hongze, Zehong Lin, and Jun Zhang. "Gi-gs: Global illumination decomposition on gaussian splatting for inverse rendering." The Thirteenth International Conference on Learning Representations.
>
> ##

---

### Official Review · Reviewer_iVxm · 2025-10-29

**Soundness:** 3
**Presentation:** 3
**Contribution:** 2
**Rating:** 4
**Confidence:** 4

**Summary:**

This paper addresses the challenge of scalable large-scale 3D reconstruction using 3D Gaussian Splatting (3DGS). The authors aim to improve both geometric accuracy and model efficiency. To this end, they propose two main technical contributions: 1) a Depth-consistent D-Normal Regularization term to enforce geometric fidelity, and 2) a Spatially adaptive Gaussian Pruning strategy to optimize the model's compactness. The authors present quantitative and qualitative comparisons against relevant baselines, supported by ablation studies, to demonstrate that their proposed method achieves more geometrically accurate reconstructions while maintaining a lightweight model.

**Strengths:**

- The paper is well-written and clearly organized, contributing to an effective presentation of the proposed method.
- The proposed method outperforms SOTA baselines across key metrics (memory consumption, geometric fidelity, novel view synthesis) and achieves visually cleaner mesh extractions on standard benchmarks (Figs. 3–4).

**Weaknesses:**

1. **Missing Related Work:** The related work section appears to omit a discussion of recent, relevant methods in large-scale 3D Gaussian Splatting, most notably CityGS-X [1]. Please discuss this work and position the current method in relation to it.
2. **Incremental Novelty:** The paper's conceptual novelty appears to be limited. The core components, such as the D-Normal regularizer and the partitioning strategy, seem to be extensions of prior art (e.g., VCR-Gaus [2] for consistency, CityGaussian/CityGS-v2 [3] for partitioning). The authors should more clearly articulate the core conceptual contributions of their work beyond the successful integration and extension of these existing ideas.
3. **Clarity of Depth Consistency:** The explanation for the depth consistency regularization requires clarification. To assess the method, please elaborate on how multi-view consistency is enforced. Specifically:
    - To which view(s) are the Gaussian depths rendered?
    - How is the depth loss calculated between these views?
4. **Motivation and Validation of SAGP:** The formulation of the SAGP component combines several known heuristics (e.g., volume, opacity). The specific weighting in Eq. 15 lacks theoretical motivation.
---
[1] Gao, Yuanyuan, et al. "Citygs-x: A scalable architecture for efficient and geometrically accurate large-scale scene reconstruction." arXiv preprint arXiv:2503.23044 (2025).

[2] Chen, Hanlin, et al. "Vcr-gaus: View consistent depth-normal regularizer for gaussian surface reconstruction." Advances in Neural Information Processing Systems 37 (2024): 139725-139750.

[3] Liu, Yang, et al. "Citygaussianv2: Efficient and geometrically accurate reconstruction for large-scale scenes." arXiv preprint arXiv:2411.00771 (2024).

**Questions:**

1. **Experimental Comparison:** Following the point on related work, please provide quantitative and qualitative comparisons against CityGS-X [1] on at least one shared dataset.
2. **Sensitivity to Priors:** How sensitive is the proposed method to the quality and noise level of the monocular priors? We request that the authors demonstrate the robustness of the confidence weighting (Eqs. 9–11), for instance by evaluating with different depth/normal estimators or by analyzing performance after injecting synthetic noise into the priors.
3. **Contribution of Depth-Consistency Term:** It is unclear if the performance gain from the depth-consistency term (Eq. 8) is due to the novel formulation itself or primarily due to the strength of the chosen prior (DepthAnything-v2 [2]). Please provide an ablation study
 that uses alternative estimators (e.g., MiDaS[3], DPT[4]) to isolate the contribution of the regularization term.
4. **SAGP Hyperparameter Analysis:** To better understand the design of the SAGP component, please provide sensitivity analyses for its key hyperparameters (e.g., the weights $\alpha, \beta, \gamma$ in Eq. 15; $\kappa$ in Eq. 14; the cell size in Eq. 13; and the pruning schedule).

---

[1] Gao, Yuanyuan, et al. "Citygs-x: A scalable architecture for efficient and geometrically accurate large-scale scene reconstruction." arXiv preprint arXiv:2503.23044 (2025).

[2] Yang, Lihe, et al. "Depth anything: Unleashing the power of large-scale unlabeled data." Proceedings of the IEEE/CVF conference on computer vision and pattern recognition. 2024.

[3] Birkl, Reiner, Diana Wofk, and Matthias Müller. "Midas v3. 1--a model zoo for robust monocular relative depth estimation." arXiv preprint arXiv:2307.14460 (2023).

[4] Ranftl, René, Alexey Bochkovskiy, and Vladlen Koltun. "Vision transformers for dense prediction." Proceedings of the IEEE/CVF international conference on computer vision. 2021.

---

> ### Author Response · Authors · 2025-11-27
> **Response to Reviewer  iVxm**
>
> We thank the reviewer for recognizing the strength and scope of our work, including its superior performance over SOTA baselines and effective presentation. We address the remaining concerns that are primarily related to clarity and technical explanation through detailed responses and additional clarifications provided below.
> ### [W1] Missing Related Work (CityGS-X)  & [Q1] Experimental Comparison
>
> Thanks. We have discussed and compared CityGS-X with ours in both Related Work and Experiments sections.
> In Sec. 2 (Related Work), similar to CityGS-X, ours also focuses on city-scale reconstruction through a parallel hierarchical 3DGS representation and progressive multi-task training. However, beyond that, we further introduce a Depth-Consistent D-Normal Regularization module, which enables holistic optimization of all Gaussian geometric parameters—especially position and rotation—by integrating rendered depth-derived normals with external depth priors. This overcomes the limitation, which struggle to update Gaussian positions effectively. Additionally, we propose a Spatially Adaptive Gaussian Pruning (SAGP) strategy that dynamically adjusts Gaussian density based on local geometric complexity and visibility, leading to more compact representations and reduced redundancy without sacrificing detail. These enhancements allow UrbanGS to achieve superior geometric accuracy and memory efficiency, as validated in our experiments on multiple urban-scale benchmarks.
> In Sec. 4 (Experiments), we compare CityGS-X with ours on the GauU-Scene dataset. As reported in Tab. 2, our method consistently achieves higher mesh reconstruction accuracy; for example, the F1 score improves from 0.456 to 0.493 on Residence and from 0.487 to 0.503 on Modern Building, with comparable or better performance on Russian Building. The qualitative comparisons in Fig. 3 further show that our reconstructions contain richer geometric details than those of CityGS-X.
>
>
> ### [W2] Incremental Novelty
>
> The idea of D-Normal regularizer and partitioning are intutive and similar concepts have introduced by recent works like VCR-Gaus (for small-scale scenes) and CityGaussian/CityGS-v2. However, we emphasize that our main contribution is the develop of D-Normal regularizer and adaptive Gaussian pruning for large-scale 3DGS, which is non-trival.
> For partitioning, we simply adjust and make it compatible with D-Normal regularizer and adaptive Gaussian pruning, and not treat it as our main contribution.
>
> Specifically, we highlight our contribution of depth-normal geometric regularizer and spatially adaptive Gaussian pruning in Common Point 1.
>
> Compared to VCR-Gaus, which also utilizes a depth-normal regularizer, our method introduces crucial innovations that enable effective large-scale urban reconstruction. The core limitation of VCR-Gaus is its strong dependence on high-quality rendered depth maps. While this assumption may hold for bounded, small-scale objects, it breaks down in complex city-scale environments. Factors such as vast viewing distances, occlusions, and sparse camera coverage lead to significant degradation and multi-view inconsistencies in the rendered depth. Consequently, the D-Normal derived from such unreliable depth becomes noisy, ultimately causing artifacts and a notable drop in geometric fidelity(See Table 2 and Figure 3).
>
> In contrast, we introduce a Depth-Consistent D-Normal Regularization framework specifically designed to address these challenges. Our key innovation lies in integrating explicit depth supervision from a robust monocular estimator—aligned with SfM scale—to directly constrain the rendered depth, which provides a stable and consistent geometric anchor that is resilient to the imperfections of 3DGS's own depth rendering in large scenes. Furthermore, we propose an adaptive confidence weighting that dynamically adjusts the influence of the depth and normal constraints based on local gradient consistency and inverse depth deviation, thereby effectively suppressing supervision from unreliable regions and preventing error propagation. This unified approach ensures robust and holistic optimization of all Gaussian parameters even in challenging large-scale settings.
>
> Compared to CityGaussian/CityGS-v2, they also use partitioning. However, they treat tiles largely independently and then rely on post-hoc merging, which may leads to boundary artifacts. Our partitioning strategy is explicitly co-designed with the depth-normal regularizer and pruning: blocks are trained under shared global geometric constraints and consistent pruning criteria, which alleviates cross-block inconsistencies and enables coherent meshes across the entire urban area.

---

> ### Author Response · Authors · 2025-11-27
> **Response to Reviewer iVxm**
>
> ### [W3] Clarity of Depth Consistency
>
> In our implementation, Gaussian depths are rendered to all training views that provide depth supervision. For each camera view $i$, we volume-render an expected depth map $\hat{D}_i$ and obtain a pseudo depth map $D_i^{\text{ext}}$ by applying DepthAnythingv2 [2] to the corresponding image and then aligning it to metric scale using sparse 3D points from COLMAP's SfM. Specifically, we compute a scale factor $s_i$ and shift $t_i$ for each view via robust least-squares fitting between the monocular depth estimates and the sparse SfM depths at valid 2D-3D correspondences, converting the relative depth predictions into scale-aligned depth. The depth loss in Eq.~(8) is defined per view as
> $$
> \mathcal{L} _ {\mathrm{id}}(u,v)
> = \bigl\lvert \hat{D} _ i^{-1}(u,v)-D _ {\mathrm{ext}}^{-1}(u,v)\bigr\rvert.
> $$
>
> Although $\mathcal{L} _ {\mathrm{id}}$ is computed separately for each view, multi-view consistency is enforced implicitly because every Gaussian has a single 3D center shared across all views; thus the same Gaussian must simultaneously satisfy the depth constraints from all cameras in which it is visible, and gradients from different views jointly update its position.
> To handle inconsistencies in the monocular pseudo depths, we reweight $\mathcal{L} _ {\mathrm{id}}$ using the confidence term $w_d$ in Eqs.~(9)--(11), which down-weights pixels with unreliable geometry based on depth-gradient agreement and normalized inverse-depth deviation.
> ### [W4] Motivation and Validation of SAGP
>
> Pruning method specifically tailored to large, city-scale environments is still underexplored. Directly applying existing Gaussian pruning schemes to urban-scale settings either keep many redundant Gaussians in uniform or distant regions (e.g., sky, far façades), leading to high memory and computation cost, or remove them too aggressively, which noticeably degrades geometric accuracy (as also reflected in Tab. 4 of the revised manuscript).
>
> This gap motivates our SAGP design: we aim to reduce redundant Gaussians while preserving those that are important for representing local geometry and appearance in large-scale scenes.
> Specifically, we introduce a simple yet robust importance score that approximates each Gaussian’s contribution to both rendering quality and geometric support under the highly non-linear 3DGS rendering process.
> Note that deriving a closed-form “theoretically optimal’’ weighting is intractable. In this case, we adopt three terms in Eq. (15) to capture complementary aspects of a primitive’s relevance: (i) the ray-intersection frequency $\phi_i$ measures how often a Gaussian contributes to visible rays and thus approximates its impact on the image formation; (ii) the opacity term $\tau_i$ reflects occlusion/visibility, suppressing nearly transparent Gaussians; and (iii) the local volume weight $w_{v,i}$, computed from percentile-based, sub-linear normalization, encodes geometric scale relative to the local density so that oversized background Gaussians are down-weighted while fine structures are preserved.
> The linear combination with coefficients $\alpha$, $\beta$, and $\gamma$ could be viewed as a standard scalarization of this multi-objective importance measure.
> Each feature is normalized to $[0,1]$, and in all experiments we simply set $\alpha =1.2$, $\beta = 1.0$, $\gamma = 0.8$, without scene-specific tuning.
> As shown in Table 3, this design already yields better F1, fewer Gaussians, and lower memory/time than alternative pruning strategies (e.g., LightGaussian pruning), indicating that the chosen weighting scheme is effective in practice. **We have clarified this motivation and the empirical robustness of SAGP in Sec. 3.3 of the revised manuscript.**

---

> ### Author Response · Authors · 2025-11-27
> **Response to Reviewer iVxm**
>
> ### [Q2] Sensitivity to Priors
> Thank you for raising this question. We follow the reviewer’s suggestion and evaluate UrbanGS with multiple depth/normal estimators. Concretely, we replace our default combination (Dav2 for depth and Dsine for normals) with three alternative depth/normal pairs built from widely used monocular predictors (Dav2 / MiDaS for depth and Dsine / GeoWizard for normals), and retrain the full pipeline on the Modern Building scene.
> The results are reported in Table D of the supplementary material. Across all four prior combinations, both rendering metrics (SSIM, PSNR, LPIPS) and geometric metrics (P, R, F1) outperform baseline siginifcantly. This implies that existing priors are all suitable for our method and our method is not sensitive to the selection of priors.
>
> More importantly, we conducted a dedicated ablation study to validate the critical role of our confidence weighting mechanism(See Table 5). When comparing the full model with Depth Anything v2 (PSNR 26.44, SSIM 0.805, F1 0.503) against the same configuration without geometry-aware confidence weighting (PSNR 26.02, SSIM 0.795, F1 0.493), we observe consistent performance degradation across all metrics. This empirical evidence confirms that our confidence weighting in Eqs. (9)-(11) is essential for effectively handling estimation uncertainties in monocular priors, rather than the robustness simply stemming from the choice of estimators. The confidence weighting mechanism enables our method to adaptively down-weight unreliable regions, making it robust to varying noise levels across different monocular estimators.
>
> ### [Q3] Contribution of Depth-Consistency Term
>
> Following the reviewer’s advice, we re-ran UrbanGS with different depth estimators while keeping our depth-consistency term (Eq.8) and all other components unchanged. Concretely, besides our default DepthAnything-v2 (Dav2) prior, we additionally use MiDaS [3] as an alternative depth estimator, leading to several depth configurations as reported in Table C of the supplementary material.
> Across all these settings, the geometric F1 score remains within a narrow range of 0.491–0.503, and rendering metrics (SSIM/PSNR/LPIPS) also show only minor variations. Importantly, in all cases UrbanGS with the depth-consistency term clearly outperforms the 3DGS baselines reported in the main paper. This indicates that the performance gain is not tied to a single strong depth prior (Dav2). Instead, the proposed depth-consistency regularization consistently improves geometry even when the depth prior is replaced by an alternative estimator such as MiDaS.

---

> ### Author Response · Authors · 2025-11-27
> **Response to Reviewer iVxm**
>
> ### [Q4] SAGP Hyperparameter Analysis
> Thank you for the suggestion. We discuss the hyperparameter in Common Point 1.
> In the revised supplementary material, we provide an explicit ablation of the three weights in Eq. (15), i.e., the coefficients $(\alpha, \beta, \gamma)$ used to combine the per-Gaussian cues. The results on the Modern Building scene are reported in Table~F. As shown in Table F, our default configuration $(1.2, 1.0, 0.8)$ achieves the best overall trade-off between rendering and geometric quality (PSNR 26.44, SSIM 0.805, LPIPS 0.157, F1 0.503). Importantly, using equal weights $(1.0, 1.0, 1.0)$ or even completely dropping one term (e.g., $(0.0, 1.0, 1.0)$, $(1.0, 0.0, 1.0)$, $(1.0, 1.0, 0.0)$) only leads to a gradual degradation, rather than catastrophic failure. This indicates that SAGP is not overly sensitive to the exact values of $(\alpha, \beta, \gamma)$.
>
> For the exponent $\kappa$ in Eq. (14), our goal is simply to apply a sub-linear compression to the local volume ratio so that extremely large Gaussians are down-weighted while preserving the ordering of Gaussians within each cell. Any choice $\kappa \in (0,1)$ has the same qualitative effect. We adopt $\kappa = 0.5$, i.e., a square-root transform, as a standard and numerically stable compromise that is widely used to compress dynamic ranges.
>
>
> Regarding the cell size in Eq.~(13), our goal is to tie the voxel resolution to the **intrinsic scale of the scene** rather than introduce a freely tunable hyperparameter. We define the characteristic cell length as
>
> $$\ell = \lambda \left(\frac{\
> {V} _ {\text{scene}}}{\mathcal{N}}\right)^{1/3},$$
>
> where $\mathcal{V} _ {\text{scene}}$ is the scene bounding-box volume and $\mathcal{N}$ is the total number of Gaussians. The term $(\mathcal{V} _ {\text{scene}}/\mathcal{N})^{1/3}$ is simply the cube root of the average volume per Gaussian, i.e., the typical inter-Gaussian spacing, so the cell size automatically adapts to the overall scene size and Gaussian density. The factor $\lambda$ is a small safety margin. We fix $\lambda = 1.2$ for all scenes so that each cell contains a sufficient number of Gaussians to yield stable local statistics.
>
> Regarding the pruning schedule, our design follows the training dynamics of 3DGS and prior practice. As shown in the pipeline, we use two stages of pruning. When constructing the coarse global Gaussian model, we apply an initial, simple pruning rule to remove obviously redundant Gaussians, reduce memory, and obtain a compact global prior for subsequent block-wise training. During block refinement, we prune at 7k, 15k, and 25k iterations (out of 30k). The 7k step is applied after the scene has roughly formed and the Gaussian distribution starts to stabilize, consistent with the behavior observed in Kerbl et al. (2023), and removes early exploratory Gaussians that no longer contribute to the final geometry. The 15k step follows the original 3DGS setting, occurring at the end of densification when the Gaussian count peaks, and is most effective for controlling model complexity and overfitting. The final pruning at 25k, inspired by LightGaussian, acts as a consolidation step near convergence, further eliminating residual redundancy and ensuring a good balance between high-fidelity reconstruction and compact, efficient rendering. We will clarify this rationale and its connection to the training dynamics in the revised manuscript.
>
> In summary, we highlight our SAGP is not sensitive to hyperparameter, and all experimental parameters in this paper are set to uniform values, with no specific adjustments made for different datasets.
>
> ##
> ### Reference
>
> [1] Gao, Y., Li, H., Chen, J., Zou, Z., Zhong, Z., Zhang, D., Sun, X., & Han, J. (2025). CityGS-X: A scalable architecture for efficient and geometrically accurate large-scale scene reconstruction. CoRR, abs/2503.23044.
>
> [2] Yang, L., et al. Depth Anything V2. Proceedings of the IEEE/CVF Conference on Computer Vision and Pattern Recognition, 2024.
>
> [3] Birkl, Reiner, Diana Wofk, and Matthias Müller. "Midas v3. 1--a model zoo for robust monocular relative depth estimation." arXiv preprint arXiv:2307.14460 (2023).

---

### Author Response · Authors · 2025-11-27
**Common Points**

## Common Points

### [Hyperparameters]
Our hyperparameters fall into two independent categories:

1.Pruning-related hyperparameters for spatially adaptive Gaussian pruning, enhancing training efficiency and memory usage without sacrificing quality.

2.Geometric regularization-related hyperparameters for depth-consistent D-Normal regularization, improving geometric accuracy by balancing directional consistency and error suppression.

Specific hyperparameter comparison experiments are as follows: the pruning weight configuration (α, β, γ) is systematically analyzed in Table F, and the geometry-aware confidence hyperparameters (γ_d, τ) are validated in the ablation studies(Table H) in the Appendix. It can be observed that our hyperparameter settings **consistently improve performance**, achieving uniform improvements across multiple datasets without requiring complex adjustment procedures.

It is particularly worth emphasizing that all experimental parameters in this paper are set to uniform values, with no specific adjustments made for different datasets. This fully highlights the **strong generalization capability** and **method robustness** of our hyperparameter settings. As shown in Table 1 and Table 2, UrbanGS consistently achieves state-of-the-art performance across urban scenes of varying scales and complexities.

### [Novelty]
Our work presents a comprehensive framework for large-scale 3D Gaussian Splatting, centered on two core contributions:

**Unified depth-normal geometric regularizer, Geometry-aware.**
Traditional D-Normal relies solely on rendered depth and is primarily applied in small-scale scenes. When extended to large-scale environments, the complexity and scale of scenes lead to insufficient rendered depth quality, resulting in low geometric accuracy (see Table 2). To address this challenge in large-scale reconstruction, we introduce pseudo-depth priors to supervise the rendered depth, and specifically propose a depth-aware confidence weighting mechanism to better align the rendered depth with these priors, thereby enhancing the effectiveness of D-Normal regularization. Extensive experiments (see Table C, Figure 4 and Figure C) demonstrate that our enhanced D-Normal achieves superior geometric consistency and reconstruction quality. Furthermore, we provide theoretical proofs showing that our method effectively updates both the position and rotation parameters of Gaussian ellipsoids (see Supplementary Material Section B and Figure E).

**Spatially adaptive Gaussian pruning.**
Traditional pruning methods, designed for small-scale or single-object scenes, typically rely on global significance metrics or fixed opacity thresholds. However, such uniform strategies become inadequate in city-scale scenarios due to extreme spatial heterogeneity and the vast number of Gaussian primitives, often leading to over-simplification of local structures or loss of fine-grained details. To overcome these limitations, we propose a spatially adaptive pruning strategy that operates locally rather than globally. Specifically, the scene is partitioned into volumetric cells where pruning decisions are made based on local geometric complexity, ray intersection frequency, and visibility-aware importance scores. Notably, our pruning module is applied progressively during both the initial coarse reconstruction and the block-wise fine-tuning stages, efficiently removing redundant Gaussians while preserving perceptually critical structures. Experimental results (see Table 4 and Figure 5) demonstrate that our method achieves a significant reduction in Gaussian counts (e.g., from 6.43M to 2.45M in the Russian scene) and training memory (from OOM to 14.4 GB), while maintaining superior rendering and geometric quality.


### [Priors]
We rigorously evaluate UrbanGS by substituting our default depth (Depth Anything V2) and normal (DSine) estimators with alternative monocular predictors (e.g., MiDaS, GeoWizard).

The results are reported in Table D of the supplementary material. Across all four prior combinations, both rendering metrics and geometric metrics outperform baseline siginifcantly. This implies that existing priors are all suitable for our method and our method is not sensitive to the selection of priors.

Ablation studies (Table 5) confirm the critical role of our confidence weighting: removing it causes consistent performance drops (e.g., PSNR 26.44→26.02, F1 0.503→0.493), proving that our mechanism (Eqs. 9-11), not the choice of estimator, is essential for handling prior uncertainties.

---

### Author Response · Authors · 2025-12-01
**To the Area Chair**

Thank you for coordinating the review process for our submission “UrbanGS: A Scalable Framework for High-Fidelity Large-Scale Scene Reconstruction.” We are grateful for the reviewers’ constructive feedback, which has helped us clarify and strengthen the presentation of our work. Below, we summarize our responses to the key points raised by each reviewer, emphasizing how our revisions have addressed their concerns.
### **Summary of Contributions**
UrbanGS introduces a comprehensive framework for city-scale 3D Gaussian Splatting with two core innovations:
1. **Depth-Consistent D‑Normal Regularization** – a geometric optimization framework designed specifically for large-scale scenes, enabling holistic updates of Gaussian position and rotation parameters and significantly improving reconstruction accuracy in city-scale environments.
2. **Spatially Adaptive Gaussian Pruning (SAGP)** – to the best of our knowledge, the first pruning method tailored for urban-scale Gaussian Splatting. It dynamically adjusts Gaussian density based on local geometric complexity and visibility, reducing memory and training time while preserving structural details.

Extensive experiments on multiple urban datasets (Mill‑19, UrbanScene3D, GauU‑Scene) demonstrate that UrbanGS consistently achieves state‑of‑the‑art performance in rendering quality, geometric fidelity, and training efficiency, using a **single set of hyperparameters** across all scenes.

---
### **Response to Reviewer 1**
Reviewer 1 raised concerns about **novelty relative to prior work (CityGS‑X, VCR‑Gaus)** and requested clarification on **depth‑consistency motivation** and **SAGP design**.

In our response, we:
- Detailed how UrbanGS advances beyond CityGS‑X by introducing depth‑aware regularization and adaptive pruning, leading to higher geometric accuracy (Table 2, Fig. 3).
- Explained that our depth‑consistent D‑Normal regularizer explicitly incorporates pseudo‑depth priors and confidence weighting to handle large‑scale depth inconsistencies, unlike VCR‑Gaus which relies solely on rendered depth.
- Provided a clear derivation of multi‑view depth consistency (Eq. 8–11) and justified SAGP’s design with local importance scoring (Eq. 15), supported by ablation studies (Table 4) showing superior compression and quality preservation.
### **Response to Reviewer 2**
Reviewer 2 raised multifaceted technical concerns, including the guarantee of multi-view consistency, the rationale behind inverse-depth weighting and confidence design, the proof of "position update along normal", and additional experiments on the GauU-Scene dataset, among others.

In our response, we systematically addressed each of these points as follows:
-  **To guarantee multi-view consistency**, we have detailed the scale-alignment of monocular depth using sparse SfM points and provided quantitative validation in Table I, reporting a high average consistency score.
-  **To justify the inverse-depth weighting and confidence design**, we conducted dedicated ablation studies. The results in Table 5 demonstrate a clear performance drop when either component is removed, confirming their critical role.
- **To substantiate the "position update along normal" proof**, we supplemented the theoretical derivation in Appendix B with an empirical visualization in Figure E, showing that our regularizer effectively drives Gaussians toward surfaces.
-  **To extend evaluation on the GauU-Scene dataset**, we have added comprehensive quantitative comparisons in Table C and qualitative mesh reconstructions in  Figure C, further demonstrating the robustness and generalizability of our method.
### **Response to Reviewer 3**
Reviewer 3 requested **hyperparameter ablation** and **additional pruning experiments**.
We have:
- Added systematic hyperparameter studies for both the geometry‑aware confidence (Table H) and the pruning score weights (Table F), confirming stable performance across a wide range of values.
- Provided a dedicated pruning ablation (Table 4) comparing SAGP against LightGaussian, showing that our method better preserves geometry while achieving greater compression and speed.
### **Conclusion**
We have thoroughly addressed each reviewer’s concerns through detailed explanations, additional experiments, and clarified presentation. UrbanGS delivers a scalable, geometry‑aware solution for urban‑scale reconstruction, consistently outperforming existing methods in rendering, geometry, and efficiency.

Thank you for your consideration.

Sincerely,
The Authors

---

### Meta-Review · Area_Chair_SSmK · 2026-01-06

**Summary:**

After reading authors' reply to authors, most of the concerns have been well resolved through extensive experiments, such as sensitivity and robustness of hyperparameters, missing comparison and ablation, selection for prior generation. Though novelty seems incremental, the authors proved that the design effectively benefits fine-grained details recovery, and significantly push the performance bound. After careful consideration, I would like to recommend the paper to be accepted as poster paper.

**Reviewer Concerns:**

Reviewer Concern: For Reviewer 1, the concern regarding missing comparsion and ablation, hyperparameter analysis, prior sensitivity has been resolved through additional experiments. For Reviewer 2, the concern regarding concept clarity, model distinction, and performance gain are resolved by thorough explanation and analysis. The robustness to prior and hyperparameters are also supported by additional experiments. For Reviewer 3, the concerns are also well resolved by aforementioned effort of rebuttal. The remaining concern isrelated novelty and motivation, which may trigger further discussion in normal rebuttal phase.

**Reviewer Scores:**

Reviewer Score: In personal opinion, after normal rebuttal phase, Reviewer 3 would maintain its score, while at least one of Reviewer 1 or Reviewer 2 would slightly increase the score, making the paper to be borderline accept

---

### Decision · Program_Chairs · 2026-01-26

Accept (Poster)